# Deep Incomplete Multi-view Learning via Cyclic Permutation of VAEs

**Xin Gao**
Fudan University
gaoxin23@m.fudan.edu.cn

**Jian Pu**[✉]
Fudan University
jianpu@fudan.edu.cn

## Abstract

Multi-View Representation Learning (MVRL) aims to derive a unified representation from multi-view data by leveraging shared and complementary information across views. However, when views are irregularly missing, the incomplete data can lead to representations that lack sufficiency and consistency. To address this, we propose **M**ulti-**V**iew **P**ermutation of Variational Auto-Encoders (MVP), which excavates invariant relationships between views in incomplete data. MVP establishes inter-view correspondences in the latent space of Variational Auto-Encoders, enabling the inference of missing views and the aggregation of more sufficient information. To derive a valid Evidence Lower Bound (ELBO) for learning, we apply permutations to randomly reorder variables for cross-view generation and then partition them by views to maintain invariant meanings under permutations. Additionally, we enhance consistency by introducing an informational prior with cyclic permutations of posteriors, which turns the regularization term into a similarity measure across distributions. We demonstrate the effectiveness of our approach on seven diverse datasets with varying missing ratios, achieving superior performance in multi-view clustering and generation tasks.

## 1 Introduction

Multi-view data are prevalent in real-world applications[1], capturing various aspects of a shared subject. Examples include observing a 3D object from multiple angles, applying diverse image feature descriptors, or reporting the same news in different languages. This offers valuable self-supervised signals that enable the extraction of meaningful patterns. Multi-View Representation Learning (MVRL) aims to maps this data into a latent space, integrating information from multiple views into a unified representation for downstream tasks like clustering, classification, and generation (Li et al., 2018). However, in practice, not all views are available for every sample, presenting the challenge of Incomplete Multi-View Representation Learning (IMVRL) (Tang et al., 2024). The incomplete data complicates the integration of information from different views, making it difficult to derive high-quality representations under varying missing ratios.

Among MVRL methods, Multimodal Variational Auto-Encoders (MVAEs) stand out for their robustness in modeling the latent distributions of multi-view data (Aguila & Altmann, 2024). Their flexibility in handling incomplete data stems from mean-based fusion strategies, such as Mixture-of-Experts (MoE) and Product-of-Experts (PoE) (Wu & Goodman, 2018; Shi et al., 2019; Sutter et al., 2021), which can accommodate varying numbers of views and have been validated by Hwang et al. (2021). Furthermore, some MVAE variants enhance inter-view consistency by incorporating data-dependent priors, which facilitate better information integration (Sutter et al., 2020; 2024) and reduce reliance on missing views (Hwang et al., 2021). Despite these advancements, the sufficiency and consistency of learned representations are still not assured in the presence of incomplete data. Samples with fewer views inherently provide less information, which hampers the integration and undermines inter-view consistency. As a result, these methods experience significant performance degradation and semantic incoherence across views as the rate of missing data increases.

---

[1]Multi-view: In this paper, we follow Hwang et al. (2021), using the term "view" broadly to refer to what other works may call view (Lin et al., 2023; Hwang et al., 2021), modality (Sutter et al., 2021; Huh et al., 2024), or any perspective describing different aspects of a common entity.

To address the challenge in IMVRL, an increasing number of studies have proposed inferring missing views through cross-generation from available ones. These methods leverage the invariant relationship between views, meaning that converting one view into another preserves the sample-specific information while altering only the style (Zhang et al., 2020; Lin et al., 2023; Cai et al., 2024; Jiang et al., 2023). This intrinsic property of multi-view data allows for a deeper extraction of information from it. Furthermore, Huh et al. (2024) observe a phenomenon of representational convergence, indicating that inter-view transformation can be easily achieved using simple mappings in an well-aligned representative space. This occurs naturally in MVAEs, where multiple encoders map different views into a latent space and align their representations by promoting inter-view consistency. Building on these insights, we aim to explicitly establish inter-view correspondences (Huang et al., 2020) in MVAEs, effectively learning and enriching the latent space with invariant relationships between views, thus enabling the inference of representations for missing views.

In this paper, we propose the **M**ulti-**V**iew **P**ermutation of VAEs (MVP), designed to learn more sufficient and consistent representations from incomplete multi-view data. MVP captures inter-view relationships by modeling correspondences between views, which enables latent variables to be transformed from one view to another. To facilitate these transformations, we apply permutations that randomly shuffle variables within each view—either through self-view encoders or cross-view correspondences. Next, we partition variables by view to preserve their invariant meanings under permutation, which helps factorize the joint posterior and derive a valid Evidence Lower Bound (ELBO) for optimization. Additionally, we propose an informational prior based on cyclic permutations of posteriors, which converts the Kullback-Leibler (KL) divergence term into a similarity measure among distributions. We validate the effectiveness of our method through experiments in multi-view clustering and generation tasks. The key contributions of this work are:

- We enhance MVAEs by modeling inter-view correspondences in the latent space to infer missing views. Our novel approach of applying permutations and partitions to the latent variable set leads to the derivation of a valid ELBO for optimization.

- We introduce an informational prior using cyclic permutations of posteriors. This results in the regularization term into a similarity measure to enhance consistency between views.

- Quantitative and qualitative results on seven diverse datasets, across different missing ratios, show that our approach learns more sufficient and consistent representations compared to other IMVRL methods and MVAEs.

## 2 RELATED WORKS

**Incomplete Multi-View Representation Learning (IMVRL)** Early IMVRL approaches addressed incomplete data by grouping available views and applying classical methods like CCA (Hotelling, 1992). DCCA (Andrew et al., 2013) introduced nonlinear representations via correlation objectives, while DCCAE (Wang et al., 2015) enhanced reconstruction with autoencoders. As missing rates increased, the need for handling incomplete information grew. Methods like DIMVC (Xu et al., 2022) projected representations into high-dimensional spaces to improve complementarity, and DSIMVC (Tang & Liu, 2022) used bi-level optimization to impute missing views. Completer (Lin et al., 2021; 2023) maximized mutual information and minimized conditional entropy to recover missing views, while CPSPAN (Jin et al., 2023) aligned prototypes across views to preserve structural consistency. ICMVC (Chao et al., 2024) proposed high-confidence guidance to enhance consistency, and DVIMC (Xu et al., 2024) introduced coherence constraints to handle unbalanced information.

**Multimodal Variational Auto-Encoders (MVAEs)** MVAEs are generative models that maximize the log-likelihood of observed data through latent variables. MVAE (Wu & Goodman, 2018) models the joint posterior using PoE (Hinton, 2002), though this may hinder unimodal posterior optimization. MMVAE (Shi et al., 2019) and mmJSD (Sutter et al., 2020) use MoE for the joint posterior, with MMVAE applying pairwise optimization for reconstructing all views, but struggling to aggregate information efficiently. mmJSD addresses this with a dynamic prior, replacing regularization with Jensen-Shannon Divergence. Sutter et al. (2021) propose Mixture-of-Product-of-Experts (MoPoE) to decompose KL divergence into $2^V$ terms, while MVTCAE (Hwang et al., 2021) introduces an information-theoretic objective using forward KL divergences. MMVAE+ (Palumbo et al., 2023) extends MMVAE by separating shared and view-peculiar information in latent subspaces and incorporating cross-view reconstructions.

**Informational Priors in VAE Formulations** Tomczak & Welling (2018) first introduced a data-dependent prior into VAE, which was later extended to multimodal VAEs for better inter-view consistency. Sutter et al. (2020) employed a dynamic prior combined with the joint posterior to define Jensen-Shannon divergence regularization. Hwang et al. (2021) used view-specific posteriors as priors, regularizing the joint posterior to ensure representations could be inferred from all views. Sutter et al. (2024) develop an MoE prior for soft-sharing of information across view-specific representations rather than simply aggregation. They relies on fusion of posteriors and enforce strict alignment, while we encourage soft consistency between views after transformations.

## 3 METHOD

In this section, we present the core components of our method, which aims to capture relationships between views in incomplete multi-view data. Our approach models inter-view correspondences in the latent space of MVAEs, enabling the inference of missing views and the generation of more sufficient and consistent representations. To facilitate transformations between views, we apply permutations to reorder the latent variables and introduce two partitions based on their encoding information (Section 3.1). We then use these partitions to factorize the posterior and derive the ELBO for optimization (Section 3.2). Finally, we define priors using the permuted variables in the regularization term. By applying cyclic permutations to the posteriors, we transform the regularization into a similarity measure, enforcing consistency across views (Section 3.3).

### 3.1 INTER-VIEW CORRESPONDENCE AND LATENT VARIABLE PARTITION

Given an incomplete multi-view dataset $\{\mathbb{X}_i\}_{i=1}^n$, where each $\mathbb{X}_i = \{x^{(v)}\}_{v \in \mathcal{I}_i}$ consists of multiple views, we denote by $\mathcal{I}_i$ a subset of the complete view indices $[L] = \{1, 2, \ldots, L\}$ (*i.e.*, $\mathcal{I}_i \subseteq [L]$). Each observed view $x^{(v)}$ is a vector in $\mathbb{R}^{d_v}$. For simplicity in derivation, we drop the subscript $i$ from $\mathcal{I}_i$ and use $\mathcal{I}$. We then encode the data from each view into a latent variable set, where each view contributes complementary information. These latent variables, $z \in \mathbb{R}^d$, are derived using encoders parameterized by $\{\phi_v\}_{v=1}^L$, following standard MVAEs.

We adopt the term "correspondences" from Huang et al. (2020), but extend it to explicitly construct a mapping channel, referred to as **inter-view correspondences**. Specifically, we introduce multiple nonlinear mappings from the $v$-th view to the $l$-th view, represented by functions $\{f_{lv}\}$, parameterized by $\{\alpha_{lv}\}$. For each pair where $v \neq l$, a unique mapping $f_{lv}$ establishes a direct relationship between the source view $v$ and the target view $l$. This allows for cross-view transformations, where information from one view informs the representation of another. The latent variables are then organized into a set $\mathcal{Z} = \{z_v^{(l)}\}_{(v,l) \in \mathcal{I} \times [L]}$, **where $z_v^{(l)}$ denotes the representation of the $l$-th view**, with subscript $v$ indicating its source view. Specifically: **(1)** If $v = l$, $z_v^{(v)}$ is directly encoded from the observed view $x^{(v)}$, following a $d$-dimensional Gaussian distribution $\mathcal{N}(z_v^{(v)}; \mu(x^{(v)}), \Sigma(x^{(v)}))$, denoted as $q(z_v^{(v)} \mid x^{(v)}; \phi_v)$. **(2)** If $v \neq l$, $z_v^{(l)}$ is transformed from $z_v^{(v)}$ using $f_{lv}$, following a Gaussian distribution $\mathcal{N}(z_v^{(l)}; f_{lv} \circ \mu(x^{(v)}), f_{lv} \circ \Sigma(x^{(v)}))$, denoted as $q(z_v^{(l)} \mid x^{(v)}; \phi_v, \alpha_{lv})$.

To organize the encoded latent variables, we construct a matrix $Z_0$, as shown in Figure 1. The diagonal elements of $Z_0$ correspond to the self-view encoding (case (1)), while the off-diagonal elements correspond to cross-view transformations (case (2)). The core idea is that variables with the same superscript $l$ encode similar information about the $l$-th view, regardless of whether they are directly encoded or transformed. Thus, even if the columns of $Z_0$ are reordered, as illustrated in the transition from $Z_0$ to $Z_1$ in Figure 1, each column continues to represent the same view, and the information encoded by elements at corresponding positions remains invariant. This observation motivates us to group the latent variables by views, leading to a partition of the set $\mathcal{Z}$. Mathematically, a **partition** $\mathcal{P}$ of a set $X$ is a collection of non-empty, mutually disjoint subsets of $X$, also known as a "set of sets". Accordingly, we define the single-view partition for $\mathcal{Z}$:

**Definition 1** (**Single-view Partition**). *A single-view partition of $\mathcal{Z}$, denoted as $\mathcal{P}_s(\mathcal{Z})$, is defined as $\{\mathcal{S}_l\}_{l=1}^L$ such that $\bigcup_{l=1}^L \mathcal{S}_l = \mathcal{Z}$. The $l$-th single-view cell $\mathcal{S}_l = \{z_v^{(l)}\}_{v \in \mathcal{I}}$ and $|\mathcal{S}_l| = |\mathcal{I}|$.*

Accordingly, each sample has a unique $\mathcal{P}_s(\mathcal{Z})$, where each single-view cell $\mathcal{S}_l$ consists of all variables with the same superscript, representing the $l$-th view. More specifically, these variables are all

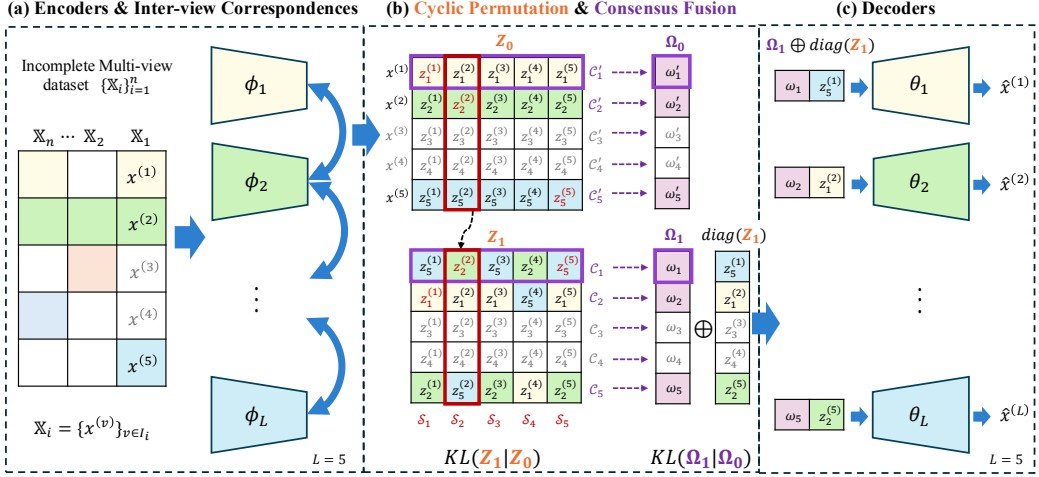

Figure 1: **Overview of our method**: (a) Incomplete multi-view data $\mathbb{X}_1$ is fed into encoders to generate the diagonal elements of matrix $Z_0$, while off-diagonal elements are derived through inter-view correspondences. (b) Latent variables are partitioned by columns for **single-view partition** $\{\mathcal{S}_i\}$ and by rows for **complete-view partition** $\{\mathcal{C}_i\}$, with each row aggregated into a consensus variable $\omega$, capturing shared information across views. A cyclic permutation within each column transforms $Z_0$ into $Z_1$, generating new partitions (See Figure 5 for transformation details.). Regularization is applied by comparing distributions at the same positions before ($Z_0$) and after ($Z_1$) permutation. (c) Each view $x^{(v)}$ is reconstructed from its latent representation $z^{(v)}$ and a consensus variable $\omega$.

drawn from the $l$-th column of matrix $Z_0$ or $Z_1$. The set $\mathcal{S}_l$ is expected to be homogeneous, meaning the distributions of the variables within it should be as close as possible.

To combine complete information across $L$ views for downstream tasks, we randomly select $L$ variables from different views in $\mathcal{Z}$. In practice, this is achieved by selecting a row from matrix $Z_0$ or $Z_1$, each representing a subset of $\mathcal{Z}$ that contains complete information from all $L$ views. This approach leads to another partition based on combinations of complete views:

**Definition 2 (Complete-view Partition).** *A complete-view partition of $\mathcal{Z}$, denoted as $\mathcal{P}_c(\mathcal{Z})$, is defined as $\{\mathcal{C}_n\}_{n\in\mathcal{I}}$ such that $\bigcup_{n\in\mathcal{I}}\mathcal{C}_n = \mathcal{Z}$. Each complete-view cell $\mathcal{C}_n$ is given by $\{z_v^{(l)}\}_{l=1,v\in\mathcal{J}_n}^L$, where the index set $\mathcal{J}_n \subseteq \mathcal{I}$ and $\bigcup_{n\in\mathcal{I}}\mathcal{J}_n = \mathcal{I}$. The size of each cell is $|\mathcal{C}_n| = L$.*

For each sample, multiple complete-view partitions satisfy Definition 2. This is evident in the matrix, where we can divide $Z_0$ by rows; each row corresponds to a complete-view cell $\mathcal{C}_n$ which consists of variables with superscripts ranging from 1 to $L$ and subscripts indicating their sources. After randomly reordering variables in each column, we obtain a new $\mathcal{P}_c(\mathcal{Z})$ by similarly dividing $Z_1$ by rows. This reordering action is rigorously described as a **permutation** of a set, which is a bijection from a set to itself. Applying permutations to each column in matrix $Z_0$ generates different $\mathcal{P}_c(\mathcal{Z})$, facilitating the selection of any complete-view combination from each row.

To aggregate the shared information from $L$ views, we introduce a **consensus variable** $\omega$, which is obtained from a complete-view cell $\mathcal{C}_n$. We assume that the first $k$ dimensions of $z$ capture information common to all views, such as categorical features, and compute the geometric mean of the $k$-dimensional marginal distributions within the set $\mathcal{C}_n$. This approach combines the $L$ complete views into a single, sharper $k$-dimensional Gaussian distribution with explicitly computed mean and covariance (Cochran, 1954), represented as $q(\omega_n \mid \mathcal{C}_n, \{x^{(v)}\}_{\mathcal{J}_n}) = \mathcal{N}_k(\omega_n; \alpha_n, \Lambda_n)$, $n \in \mathcal{I}$. Thus, a complete-view partition $\mathcal{P}_c(\mathcal{Z})$ with $|\mathcal{I}|$ cells produces a set $\boldsymbol{\Omega} = \{\omega_n|\omega_n \sim q(\omega_n \mid \mathcal{C}_n, \{x^{(v)}\}_{\mathcal{J}_n}), \mathcal{C}_n \in \mathcal{P}_c(\mathcal{Z}), n \in \mathcal{I}\}$. The set $\boldsymbol{\Omega}$ should also be homogeneous, meaning that any combination of $L$ views fused in this way should encode consistent information.

By defining these two partitions, we enable the factorization of the posterior in the following sections, forming the foundation for our learning objective.

### 3.2 Posterior Factorization and the Derivation of ELBO

Given the latent variables $\boldsymbol{\mathcal{Z}}$ and $\boldsymbol{\Omega}$, our goal is to infer their posterior distribution $p(\boldsymbol{\mathcal{Z}}, \boldsymbol{\Omega} \mid \{x^{(v)}\}_{\mathcal{I}})$. However, this exact posterior involves an intractable integral, so we approximate it with a variational posterior $q(\boldsymbol{\mathcal{Z}}, \boldsymbol{\Omega} \mid \{x^{(v)}\}_{\mathcal{I}})$. For a given complete-view partition $\mathcal{P}_c(\boldsymbol{\mathcal{Z}}) = \{\boldsymbol{\mathcal{C}}_n\}_{n \in \mathcal{I}}$, we assume that the density factorizes as follows:

$$q(\boldsymbol{\mathcal{Z}}, \boldsymbol{\Omega} \mid \{x^{(v)}\}_{\mathcal{I}}) \triangleq \prod\nolimits_{n \in \mathcal{I}} q(\boldsymbol{\omega}_n \mid \boldsymbol{\mathcal{C}}_n, \{x^{(v)}\}_{\mathcal{J}_n}) q(\boldsymbol{\mathcal{C}}_n \mid \{x^{(v)}\}_{\mathcal{J}_n})$$

$$= \prod\nolimits_{n \in \mathcal{I}} \left[ q(\boldsymbol{\omega}_n \mid \boldsymbol{\mathcal{C}}_n, \{x^{(v)}\}_{\mathcal{J}_n}) \prod\nolimits_{l=1, v \in \mathcal{J}_n}^{L} q(\boldsymbol{z}_v^{(l)} \mid x^{(v)}) \right].$$

To optimize this factorized posterior, we minimize the KL divergence between the true posterior and the variational posterior, expressed as:

$$KL\left[ q(\boldsymbol{\mathcal{Z}}, \boldsymbol{\Omega} \mid \{x^{(v)}\}_{\mathcal{I}}) \parallel p(\boldsymbol{\mathcal{Z}}, \boldsymbol{\Omega} \mid \{x^{(v)}\}_{\mathcal{I}}) \right] = \log p(\{x^{(v)}\}_{\mathcal{I}}) - \mathcal{L}_{\text{ELBO}}(\{x^{(v)}\}_{\mathcal{I}}),$$

where $\mathcal{L}_{\text{ELBO}}(\{x^{(v)}\}_{\mathcal{I}})$ is the Evidence Lower Bound (ELBO) of the log-likelihood of incomplete multi-view data $\{x^{(v)}\}_{\mathcal{I}}$. Maximizing the ELBO effectively minimizes the KL divergence, thereby improving the approximation of the true posterior. Next, we use decoders parameterized by $\{\theta_v\}_{v=1}^{L}$ to reconstruct the observations. Each view $x^{(n)}$ is reconstructed using the latent variable $z_*^{(n)}$ (represent the $n$-th view) and a consensus variable $\omega_n$. The likelihood can be written as:

$$p(x^{(n)} | \boldsymbol{\mathcal{C}}_n, \omega_n) = p\left( x^{(n)} | \boldsymbol{\mathcal{C}}_n \cap \boldsymbol{\mathcal{S}}_n, \omega_n; \theta_n \right),$$

where $\boldsymbol{\mathcal{C}}_n \cap \boldsymbol{\mathcal{S}}_n = z_*^{(n)}$ is the diagonal element of $Z_1$, with its source $*$ determined by the permutation. Then the ELBO is expressed as (with detailed derivations in Appendix A.4):

$$\mathcal{L}_{\text{ELBO}}(\{x^{(v)}\}_{\mathcal{I}}) = \sum_{n \in \mathcal{I}} \mathbb{E}_{q(\boldsymbol{\mathcal{C}}_n, \omega_n | \{x^{(v)}\}_{\mathcal{J}_n})} \left[ \log p(x^{(n)} \mid \boldsymbol{\mathcal{C}}_n \cap \boldsymbol{\mathcal{S}}_n, \omega_n) \right]$$

$$- \sum_{l=1}^{L} \sum_{v \in \mathcal{I}} KL\left[ q(z_v^{(l)} \mid x^{(v)}) \parallel p(z_v^{(l)}) \right] - \sum_{n \in \mathcal{I}} KL\left[ q(\omega_n \mid \boldsymbol{\mathcal{C}}_n, \{x^{(v)}\}_{\mathcal{J}_n}) \parallel p(\omega_n) \right]. \tag{1}$$

The ELBO in Eq. (1) consists of three terms. The first term represents the reconstruction loss for all observed views, ensuring that the variable $z_*^{(n)} = \boldsymbol{\mathcal{C}}_n \cap \boldsymbol{\mathcal{S}}_n$ encodes information of $n$-th view, while $\omega$ captures shared aspects. By applying permutations to obtain a new $\mathcal{P}_c(\boldsymbol{\mathcal{Z}})$, we can derive another $\mathcal{L}_{\text{ELBO}}$. This randomness in permutation results in $z_*^{(n)}$ with different source views $*$, facilitating both **self-view reconstruction** and **cross-view generation**. We can even use a convex combination of different $\mathcal{L}_{\text{ELBO}}$, discussed in Appendix A.4. Different partitions produce varying $\omega$ that serve the same role, ensuring that the first $k$ dimensions capture shared features across views. The remaining two terms are regularization terms for $z$ and $\omega$, further discussed in Section 3.3.

### 3.3 Prior Setting using Cyclic Permutations of Posteriors

The regularization terms in the ELBO for $z$ and $\omega$, when their priors are properly set, support our learning objective of establishing inter-view correspondences in MVAEs. In the first term for $z$, the outer summation spans all single-view cells $\boldsymbol{\mathcal{S}}_l$, while the inner summation includes all $l$-th view latent variables $z$ in $\boldsymbol{\mathcal{S}}_l$, or equivalently, all variables in the $l$-th column of matrices $Z_0$ or $Z_1$. With well-established inter-view correspondences, the $l$-th view variables should encode similar information and have distributions that are as close as possible, regardless of their sources.

In Section 3.1, we introduced the idea of permuting the columns of matrix $Z_0$ to generate different complete-view partitions. When these permutations follow a cyclic structure, the permuted posteriors can be used as informational priors. For example, if we have three distributions $P_1, P_2, P_3$, we can set the prior for $P_1$ as $P_3$, for $P_2$ as $P_1$, and for $P_3$ as $P_2$. As these pairs converge, the distributions become more similar due to the cyclic nature of the permutation ($P_1 \Rightarrow P_3 \Rightarrow P_2 \Rightarrow P_1$). This **cyclic permutation** of indices ensures that the latent variables in each single-view cell become more homogeneous. Cyclic permutations can be efficiently generated before training using Sattolo's Algorithm with time complexity $O(N)$ (Wilson, 2005), defined as follows:

Table 1: **Summary of the seven multi-view datasets**. These benchmarks exhibit diverse characteristics, with the last two image datasets enabling better observation through visualization.

| Dataset | #View | Type | Dimension | #Sample | #Class |
|---|---|---|---|---|---|
| Handwritten | 6 | Fou, Fac, Kar, Zer, Pix, Mor | 76, 216, 64, 47, 240, 6 | 2000 | 10 |
| CUB | 2 | Image, Text | 1024, 300 | 600 | 10 |
| Scene15 | 3 | GIST, PHOG, LBP | 20, 59, 40 | 4485 | 15 |
| Reuters | 5 | Text: Eng., Fr., Ger., It., Sp. | 10, 10, 10, 10, 10 | 18758 | 6 |
| SensIT Vehicle | 2 | Acoustic, Seismic | 50, 50 | 98528 | 3 |
| PolyMNIST | 5 | RGB Image | 3×28×28 | 70000 | 10 |
| MVShapeNet | 5 | RGB Image | 3×64×64 | 25155 | 5 |

**Definition 3** (**Cyclic Permutation** (Gross, 2016)). *A cyclic permutation of a set $X$ is a bijection $\sigma : X \to X$. For any $x \in X$, $\sigma^{|X|}(x) = x$ and $\sigma^k(x) \neq x$ for all $k < |X|, k \in \mathbb{N}_+$.*

In our model, for each single-view cell $\mathcal{S}_l$, we define the prior for $z_v^{(l)}$ as a cyclic permutation of its corresponding posterior. Specifically, we compute the KL divergence between the distributions at the same positions in $Z_0$ and $Z_1$ (before and after permutation). This leads to a new measure of similarity within each single-view cell, which we define as *Permutation Divergence*, aimed at minimizing the heterogeneity among distributions.

**Definition 4** (**Permutation Divergence**). *Let $N \geq 2$ be a fixed natural number. Given a cyclic permutation $\sigma$ on the index set $[N]$ and a set $\mathcal{P}$ of probability distributions on the same measure space, the Permutation Divergence of order $N$ is a mapping $d$ from $\mathcal{P}^N$ to the extended real line $\mathbb{R} \cup \{+\infty\}$, defined as follows for any $P_1, P_2, \ldots, P_N \in \mathcal{P}$:*

$$d(P_1, P_2, \ldots, P_N; \sigma) = \sum\nolimits_{i=1}^{N} KL[P_i \parallel P_{\sigma(i)}].$$

The proof that Permutation Divergence is a valid similarity measure is provided in Appendix A.3. The key property is that $d(P_1, P_2, \ldots, P_N) = 0$ if and only if $P_1 = P_2 = \cdots = P_N$. Minimizing the Permutation Divergence ensures that the distributions of the latent variables in each single-view cell become as similar as possible, whether they are self-encoded or cross-transformed. This reflects how well the model captures inter-view correspondences, a property we refer to as **Inter-View Translatability**. This approach also enforces a form of soft consistency in the latent space, which encourages easier transformation between views rather than strict alignment.

The second regularization term in Eq. (1) targets the consensus variable $\omega$, deterministically derived from $z$ via the geometric mean of marginal distributions in $\mathcal{C}_n$. Serving as a central anchor across $L$ views, $\omega$ actually acts as an additional regularizer for $z$. Each $\omega_n$ captures shared information across views and should be similar, with its prior set as the posterior fused from another complete-view cell $\mathcal{C}'_n$. Minimizing it ensures the similarity of all $\omega_n$, a property we refer to as **Consensus Concentration**. For visualizations of their impacts, please refer to Appendix A.5.

## 4 EXPERIMENTS

We extensively evaluated the proposed method across seven diverse multi-view datasets, summarized in Table 1. These datasets encompass a variety of view types with different dimensions, originating from diverse sensors or descriptors, as well as real-world perspectives captured from different angles. PolyMNIST (Sutter et al., 2021) consists of five images per data point, all sharing the same digit label but varying in handwriting style and background. ShapeNet is a large-scale repository of 3D CAD models of objects (Chang et al., 2015). For each object, we rendered five images from viewpoints spaced 45 degrees apart around the front of the object. Then we selected five representative categories to create a multi-view dataset called MVShapeNet. The missing patterns are predetermined and saved as masks before training. Specifically, for each missing rate $\eta = \{0.1, 0.3, 0.5, 0.7\}$, we randomly select $\eta \times \text{len(dataset)}$ samples to be incomplete, ensuring that each incomplete sample retains at least one view while missing at least one. For more details on the generation of missing patterns and masks, please refer to Appendix C.2.

In Section 4.1, we perform clustering in the latent space and compare our method to eight state-of-the-art IMVRL approaches. The results, consistent across various missing ratios, underscore the

Table 2: **Clustering results of nine methods on five multi-view datasets with missing rates** $\eta = \{0.1, 0.3, 0.5, 0.7\}$. The first and second best results are indicated in **bold red** and **blue**, respectively. Each experiment was run five times, with the means reported here due to space limit. The complete table, including standard deviations, is provided in Table 3 from Appendix B.

| | Missing Rate | 0.1 | | | 0.3 | | | 0.5 | | | 0.7 | | |
|---|---|---|---|---|---|---|---|---|---|---|---|---|---|
| | Metrics | ACC↑ | NMI↑ | ARI↑ | ACC↑ | NMI↑ | ARI↑ | ACC↑ | NMI↑ | ARI↑ | ACC↑ | NMI↑ | ARI↑ |
| Handwritten | DCCA | 73.83 | 71.10 | 55.13 | 68.02 | 64.93 | 44.76 | 63.25 | 60.27 | 38.93 | 59.07 | 56.44 | 33.58 |
| | DCCAE | 73.61 | 72.22 | 54.61 | 68.12 | 65.76 | 43.42 | 63.36 | 60.72 | 37.32 | 59.31 | 56.89 | 32.55 |
| | DIMVC | 89.13 | 80.06 | 77.96 | 85.24 | 74.67 | 70.83 | 82.76 | 71.17 | 66.81 | 79.66 | 68.94 | 63.12 |
| | DSIMVC | 81.27 | 79.47 | 71.59 | 81.82 | 80.27 | 73.36 | 81.39 | 79.23 | 71.88 | 77.38 | 74.80 | 66.84 |
| | Completer | 82.18 | 77.73 | 68.92 | 78.70 | 69.08 | 58.14 | 74.73 | 67.49 | 50.21 | 68.86 | 62.41 | 41.06 |
| | CPSPAN | 80.30 | 78.43 | 71.84 | 79.80 | 79.08 | 72.28 | 84.64 | 80.23 | 75.34 | 83.90 | 80.60 | 75.25 |
| | ICMVC | 88.86 | 82.19 | 79.51 | 80.95 | 74.53 | 68.15 | 73.83 | 67.95 | 58.72 | 67.48 | 61.99 | 48.66 |
| | DVIMC | 87.89 | 83.51 | 79.66 | 85.36 | 82.82 | 78.50 | 85.01 | 82.96 | 78.50 | 81.66 | 80.78 | 74.85 |
| | **MVP(Ours)** | 90.55 | 87.08 | 84.46 | 88.69 | 84.86 | 81.59 | 90.76 | 85.44 | 83.53 | 86.74 | 82.56 | 78.75 |
| CUB | DCCA | 58.93 | 59.59 | 40.87 | 55.65 | 54.23 | 35.63 | 48.60 | 46.71 | 27.88 | 42.35 | 39.42 | 21.77 |
| | DCCAE | 57.27 | 61.04 | 43.09 | 53.43 | 54.59 | 36.48 | 47.21 | 47.05 | 28.37 | 41.52 | 40.50 | 22.66 |
| | DIMVC | 66.03 | 61.70 | 48.96 | 57.20 | 56.00 | 41.25 | 60.65 | 55.75 | 42.81 | 56.08 | 51.07 | 36.65 |
| | DSIMVC | 72.93 | 67.82 | 55.89 | 66.83 | 61.78 | 47.71 | 68.37 | 61.55 | 48.21 | 67.33 | 59.89 | 46.31 |
| | Completer | 52.97 | 65.47 | 45.98 | 60.73 | 68.88 | 52.78 | 51.90 | 61.84 | 45.11 | 19.43 | 17.37 | 0.73 |
| | CPSPAN | 58.77 | 62.27 | 45.35 | 61.30 | 64.21 | 48.93 | 60.07 | 64.18 | 46.42 | 58.60 | 62.16 | 45.23 |
| | ICMVC | 29.23 | 38.31 | 21.55 | 24.33 | 25.74 | 14.17 | 22.90 | 19.87 | 10.52 | 19.43 | 16.10 | 7.79 |
| | DVIMVC | 44.53 | 41.83 | 23.78 | 43.37 | 45.18 | 28.29 | 39.57 | 34.39 | 20.52 | 39.47 | 36.71 | 22.41 |
| | **MVP(Ours)** | 78.67 | 77.67 | 66.73 | 74.97 | 73.09 | 61.35 | 74.20 | 71.40 | 59.11 | 66.53 | 63.11 | 50.59 |
| Scene15 | DCCA | 38.22 | 41.20 | 19.89 | 36.16 | 39.46 | 17.12 | 34.05 | 37.26 | 14.48 | 30.84 | 33.93 | 12.60 |
| | DCCAE | 39.46 | 42.08 | 20.36 | 36.73 | 39.80 | 17.06 | 34.49 | 37.66 | 14.57 | 31.16 | 34.21 | 12.64 |
| | DIMVC | 42.51 | 41.53 | 24.45 | 40.37 | 38.57 | 20.84 | 40.17 | 35.95 | 20.59 | 36.01 | 32.57 | 16.29 |
| | DSIMVC | 29.43 | 30.38 | 14.86 | 31.38 | 32.54 | 16.29 | 27.24 | 28.68 | 13.38 | 28.42 | 29.09 | 13.85 |
| | Completer | 37.00 | 41.89 | 23.60 | 40.04 | 42.41 | 24.22 | 36.64 | 38.99 | 19.70 | 35.37 | 37.05 | 17.58 |
| | CPSPAN | 42.69 | 38.79 | 24.56 | 43.21 | 39.42 | 24.94 | 43.44 | 39.19 | 24.96 | 42.53 | 38.41 | 24.38 |
| | ICMVC | 43.88 | 40.03 | 25.80 | 43.14 | 38.06 | 24.74 | 37.96 | 33.45 | 20.34 | 36.70 | 35.80 | 18.35 |
| | DVIMC | 45.16 | 45.06 | 28.64 | 43.68 | 42.32 | 26.68 | 41.13 | 39.58 | 25.03 | 39.59 | 36.66 | 21.56 |
| | **MVP(Ours)** | 45.70 | 43.77 | 27.90 | 45.81 | 42.54 | 27.53 | 45.28 | 41.84 | 26.80 | 43.14 | 39.53 | 24.68 |
| Reuters | DCCA | 47.66 | 23.93 | 15.46 | 46.28 | 20.62 | 12.71 | 44.10 | 22.63 | 11.04 | 43.36 | 22.90 | 10.03 |
| | DCCAE | 42.70 | 23.84 | 7.59 | 43.71 | 26.07 | 8.15 | 42.32 | 24.30 | 6.80 | 41.32 | 23.11 | 5.90 |
| | DIMVC | 48.83 | 28.94 | 25.78 | 50.54 | 29.86 | 26.89 | 48.51 | 27.29 | 24.74 | 46.94 | 25.79 | 23.24 |
| | DSIMVC | 51.26 | 35.56 | 28.21 | 51.33 | 34.88 | 26.61 | 50.78 | 36.85 | 28.27 | 47.12 | 33.57 | 25.51 |
| | Completer | 41.08 | 21.38 | 7.92 | 40.56 | 22.48 | 10.32 | 41.77 | 20.41 | 9.80 | 42.27 | 22.47 | 11.51 |
| | CPSPAN | 38.35 | 14.35 | 10.94 | 38.51 | 13.11 | 10.47 | 38.21 | 11.80 | 11.30 | 37.86 | 12.03 | 10.16 |
| | ICMVC | 54.01 | 36.52 | 29.44 | 51.09 | 30.71 | 25.66 | 47.59 | 28.43 | 23.56 | 47.67 | 26.83 | 22.14 |
| | DVIMC | 44.06 | 16.08 | 15.21 | 43.06 | 10.84 | 11.77 | 35.37 | 5.14 | 4.98 | 32.18 | 3.02 | 3.15 |
| | **MVP(Ours)** | 57.83 | 37.25 | 32.20 | 55.70 | 37.02 | 31.35 | 53.67 | 35.43 | 30.24 | 55.16 | 36.00 | 30.66 |
| SensIT Vehicle | DCCA | 57.11 | 11.60 | 14.26 | 57.76 | 14.46 | 16.62 | 53.89 | 11.01 | 12.79 | 50.69 | 8.47 | 9.75 |
| | DCCAE | 57.93 | 12.84 | 15.28 | 60.32 | 19.42 | 22.46 | 54.08 | 13.32 | 15.40 | 51.33 | 10.31 | 11.81 |
| | DIMVC | 59.72 | 17.31 | 21.82 | 62.38 | 23.18 | 27.93 | 61.09 | 22.08 | 26.21 | 60.57 | 21.36 | 25.44 |
| | DSIMVC | 69.82 | 33.40 | 34.88 | 69.24 | 32.95 | 33.50 | 68.05 | 31.49 | 31.56 | 66.54 | 30.08 | 29.73 |
| | Completer | 52.63 | 5.33 | 3.72 | 55.59 | 12.09 | 11.29 | 55.09 | 13.96 | 12.52 | 56.37 | 14.77 | 14.66 |
| | CPSPAN | 63.48 | 28.43 | 32.32 | 64.03 | 28.10 | 32.33 | 65.47 | 28.62 | 32.25 | 64.16 | 28.60 | 31.28 |
| | ICMVC | 71.50 | 34.53 | 36.41 | 70.79 | 32.95 | 33.63 | 67.80 | 29.11 | 29.36 | 54.11 | 19.39 | 18.93 |
| | DVIMC | 69.48 | 30.41 | 34.98 | 69.58 | 30.31 | 35.26 | 67.89 | 29.27 | 34.00 | 61.91 | 25.84 | 28.59 |
| | **MVP(Ours)** | 72.08 | 34.81 | 41.10 | 71.28 | 33.57 | 39.76 | 70.21 | 32.05 | 38.06 | 68.87 | 30.08 | 36.23 |

structural robustness and informativeness of our learned representations. To further evaluate the effectiveness of our posterior and prior settings within the VAE framework, Section 4.2 compares our method with six MVAEs using two image datasets, PolyMNIST and MVShapeNet. Through multi-view clustering and generation tasks on different missing combinations, we demonstrate that our method learns representations with greater sufficiency and consistency. Additionally, an ablation study on the use of permutation, permutation types, and prior settings is provided in Appendix B.3, which confirms that our cyclic approach delivers the best results.

## 4.1 ENHANCED PERFORMANCE FOR INCOMPLETE MULTI-VIEW CLUSTERING

To evaluate the ability of our method to handle incomplete multi-view data, we first assess clustering performance under various missing ratios, following Zhang et al. (2020); Lin et al. (2023); Cai et al. (2024). We apply K-means clustering to the consensus representation $\omega$ and compare our method with eight IMVRL approaches: **DCCA**, **DCCAE**, **DIMVC**, **DSIMVC**, **Completer**, **CPSPAN**, **ICMVC**, and **DVIMC** (see Section 2 and Table 9 in Appendix D.1 for their modeling details). Clustering performance is measured by Accuracy (ACC), Normalized Mutual Information (NMI), and Adjusted Rand Index (ARI) in previous works.

The experimental results in Tables 2 and 3 show that our method consistently achieved the best (**bold red**) or second-best (**bold blue**) performance across various missing ratios and datasets. This highlights the robustness of our method on both large and small datasets with varying class numbers. In contrast, other methods fluctuated significantly across datasets. Notably, on smaller datasets like Handwritten, with six views provided rich self-supervised information, our method maintained strong performance even as missing rates $\eta$ increased. On CUB, where image-text modality disparity is larger, our approach excelled at moderate missing rates ($\eta = 0.1$, ACC: 78.67% vs. 72.93%; $\eta = 0.3$, ACC: 74.97% vs. 66.83%) by better integrating complementary information from different views. However, as $\eta$ rose, the limited views and small sample size led to a faster decline in performance than on other datasets. Still, our method stayed competitive with DSIMVC, which uses bi-level optimization to impute missing views, far outperforming the VAE-based DVIMC. On SensIT Vehicle, with also two views but a larger sample size, our method experienced a smaller performance drop and maintained the best results even at higher missing ratios. The Reuters, with its large size but more views, more clearly highlighted our method's advantage, achieving an 8.04% lead in ACC over the second-best method at $\eta = 0.7$. For datasets with more classes, like Scene15 and Handwritten, our method performed comparably to DVIMC, which uses a Gaussian Mixture for explicit class modeling in the VAE. However, as $\eta$ increased, DVIMC struggled with incomplete information due to missing views, while our method mitigated this by inferring complete-view information, maintaining robustness.

## 4.2 COMPARING WITH OTHER MVAEs USING TWO IMAGE DATASETS

In this section, we compare our method with six other MVAEs that utilize different posterior and prior modeling techniques: **mVAE** (Wu & Goodman, 2018), **mmVAE** (Shi et al., 2019), **MoPoE** (Sutter et al., 2021), **mmJSD** (Sutter et al., 2020), **MVTCAE** (Hwang et al., 2021), and **MMVAE+** (Palumbo et al., 2023). These models can naturally adapt to incomplete scenarios because their mean-based fusion can accommodate any number of views, as validated by Hwang et al. (2021). We perform experiments on two tasks: **multi-view clustering** and **generation**, to demonstrate that our learned representation is able to extract more sufficient information from multiple views and infer missing views from incomplete observations while maintaining consistent semantics. We conduct our evaluation using two image datasets, PolyMNIST and MVShapeNet.

### 4.2.1 POLYMNIST: PRESERVING CONSISTENT SEMANTICS ACROSS DIVERSE STYLES

For the PolyMNIST dataset, the shared information across its five views is the digit ID, while view-specific details include handwriting styles and backgrounds. Although the digit is present in each view, it can be obscured or unclear in some images, making it crucial to aggregate complementary information from all views for accurate recognition. We use the original split with 60K tuples for training and 10K for testing, All models are trained on incomplete observations ($\eta = 0.5$), with 50% of samples having 1 to 4 views missing. We evaluate model performance on the testing data across all possible incomplete view combinations, totaling $C_5^1 + C_5^2 + C_5^3 + C_5^4 = 30$ cases.

**Evaluation protocol** At test time, given the incomplete subset $\{x^{(v)}\}_{\mathcal{I}}$, we extract the representation using $|\mathcal{I}|$ encoders and evaluate its quality. We perform K-means clustering directly and use Normalized Mutual Information (NMI) as the performance metric. Next, we generate all views $\{x^{(v)}\}_{[L]}$ using the corresponding decoders. To assess consistent semantics across views, we measure coherence accuracy by feeding the generated views into a pretrained CNN-based classifier and checking if the predictions match the labels of the given subsets. Finally, we use the Structural Similarity Index Measure (SSIM) to compare the similarity between the reconstructions and the ground truth. All results are averaged across subsets of the same size.

**Results** The left plot in Figure 2 shows that our method consistently outperforms others in clustering across various incomplete scenarios. As the number of missing views increases, PoE- and MoE-based fusion methods experience sharp performance declines due to severe incomplete information. In contrast, only our method and MVTCAE maintain high levels of structural information. Our approach explicitly establishes inter-view correspondences to compensate for missing information, encoding different views into a latent space that facilitates easier transformations between them. MVTCAE penalizes latent information that cannot be inferred from other views to retain only highly correlated details. Both methods enforce a form of consistency in the representation, ensuring that the aggregated information is less affected by the presence of missing views.

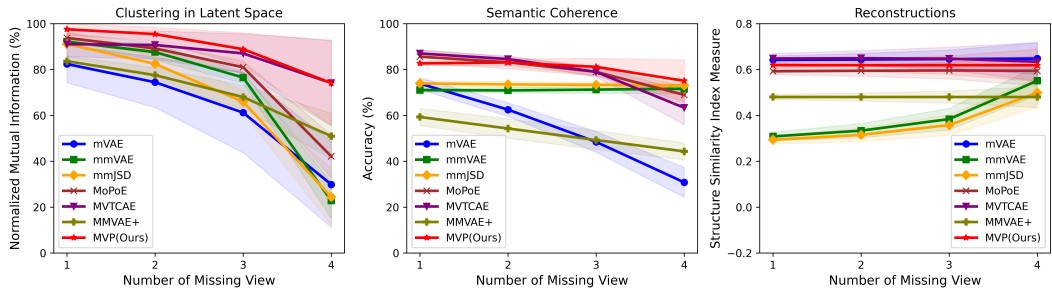

Figure 2: **Quantitative results on the PolyMNIST dataset compared to six MVAEs.** Evaluations were conducted on all incomplete subsets of the testing set, averaged across same-sized subsets.

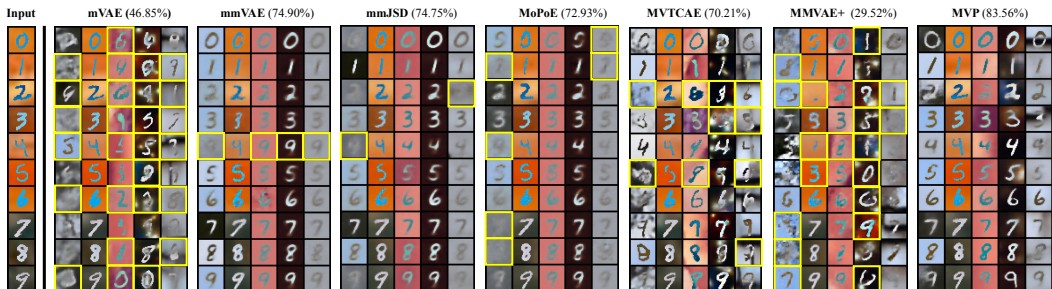

Figure 3: **Multi-view sample generation conditioned on view 2.** The leftmost column shows input images of view 2, randomly selected from digit classes 0 to 9. The following columns display multi-view samples (five views per sample) generated by various models. Ideally, the conditional generated digits should match the input digit, with yellow boxes highlighting inconsistencies. Accuracy scores, shown in parentheses, are derived from pre-trained classifiers on the generated images.

The middle plot in Figure 2 shows the semantic coherence results. Our method exhibits a smaller performance drop and surpasses others as missing views increase. In contrast, other models show an obvious decline when up to four views are missing. Perversely, MMVAE and mmJSD slightly improve as more views are lost due to MoE's limitations in aggregating information (Hwang et al., 2021), which favors single-view identification. Figure 3 illustrates a task where only view 2 was provided, with representations extracted from incomplete observations to generate full views. Our method achieved 83.56% accuracy in maintaining semantic consistency with the input, significantly outperforming other models. In contrast, other methods showed notable inconsistencies across the five-view tuples, highlighted by the yellow boxes. mVAE struggled to maintain consistent semantics across views due to the precision miscalibration of each view caused by PoE fusion (Shi et al., 2019). MoE-based models like mmVAE, mmJSD, and MoPoE maintained some semantic coherence but failed with more challenging samples, producing blurry backgrounds. MVTCAE generated clearer, more varied backgrounds but still showed semantic inconsistencies in several samples. Although designed to retain highly correlated information between views, missing views made it harder to maintain consistency, especially with only one view available. MMVAE+ produced clearer backgrounds than mmVAE by separating shared and view-specific subspaces, but sampling missing-view information from auxiliary priors caused severe category confusion.

The right plot in Figure 2 shows the SSIM results. Also, mmVAE and mmJSD exhibit improved reconstruction quality as the number of missing views increases—a counterintuitive trend, yet consistent with theirs in semantic coherence. MMVAE+ performs moderately, likely because we used the same network structure for all models rather than its original ResNet architecture, suggesting it may overly rely on powerful decoders for good generation. Since SSIM primarily reflects the quality of dominant backgrounds in the PolyMNIST dataset, our method is the only one that consistently balances high semantic coherence with diverse background styles.

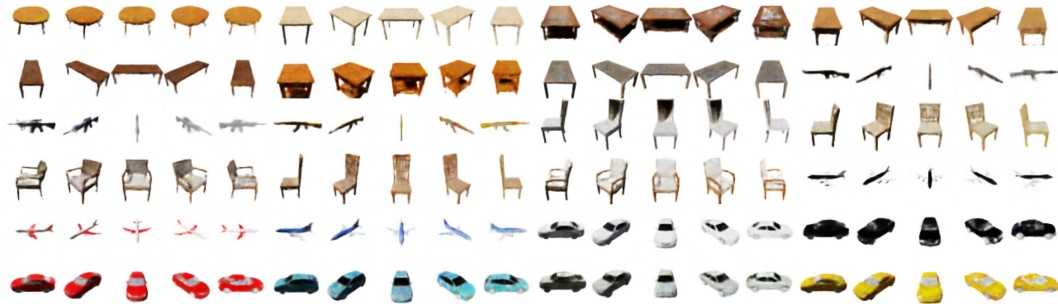

Figure 4: **Multi-view samples generated by our method on the MVShapeNet dataset.** Categories include table, chair, car, airplane, and rifle, with each sample consisting of five views from different angles. The model was trained with missing rate $\eta = 0.5$ and tested with only view 5 available.

### 4.2.2    MVShapeNet: Capturing Detailed Information from Various Angles

We further evaluate our method on the MVShapeNet dataset. Unlike PolyMNIST, where views share a few common pixels depicting the same digit against various background styles, MVShapeNet presents a smaller inter-view gap and greater consistency due to its uniform white backgrounds with the same object captured from different real-world angles. We use an 80:20 train-test split and apply the same experimental settings as in Section 4.2.1.

In this setting, MVAEs are less prone to semantic inconsistencies observed with PolyMNIST, meaning the generated object generally matches the input. However, we expect the learned representations to preserve more detail such as furniture hollowing, textures, and lightnings, which remains a challenge under high missing ratios. To evaluate this, we compare our method with other MVAEs at missing rates of $\eta = 0.1$ and $0.5$, testing across all possible incomplete combinations. For quantitative evaluation, we use average SSIM to evaluate the basic structure of generated images. Additionally, we pretrain two CNN-based classifiers on all views to evaluate whether the decoded images accurately capture object categories and perspective angles. As demonstrated in Table 4 of Appendix B.1, our method consistently performs well, regardless of whether it is trained at high or low missing rates and tested on any incomplete combination. In contrast, other methods either show a dramatic performance drop as the number of missing views increases or rely on memorizing incomplete samples without effectively aggregating complementary information from additional views. As shown in Figure 4, when only view 5 is provided, MVP leverages inter-view correspondences to transform latent representations and successfully infer the missing views. Further visualizations in Figure 10 illustrates that other models tend to produce blurry reconstructions with a lack of detail or unclear perspective angles. In contrast, our method clearly infers more accurate details in missing views, such as the placement of table legs at different angles, changes in light and shadow, and hollowed-out armrests on chairs. All these results suggest that our method learns representation with more sufficient and consistent information from incomplete multi-view data.

## 5    Conclusion

In this paper, we presented Multi-View Permutation of VAEs (MVP), a novel framework to address the challenges of incomplete multi-view learning. By explicitly modeling inter-view correspondences in the latent space, MVP effectively captured invariant relationships between views. We derived a valid ELBO for efficient optimization by applying permutation and partition operations to the latent variable set. Notably, these operations on multi-view representations are not limited to the VAE framework and can be extended into non-generative models. Additionally, the introduction of an informational prior using cyclic permutations of posteriors resulted in regularization terms with both practical meanings and theoretical guarantees. Extensive experiments on seven diverse datasets demonstrated the robustness and superiority of MVP over existing methods, particularly in scenarios with high missing ratios. These findings underscore its potential to reveal more informative latent spaces and fully unlock the capability of MVAEs to handle incomplete data.

## ACKNOWLEDGMENTS

Special thanks to Yuxin Li, Hangqi Zhou, Hanru Bai, An Sui, Fuping Wu, Yuanye Liu, Yibo Gao, and Ruofeng Mei for their invaluable feedback on the manuscript.

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

# Appendices

## A    THEORETICAL ANALYSIS

In this section, we present a comprehensive theoretical analysis of the proposed method, offering additional details to complement the main text.

### A.1    PARTITION AND PERMUTATION OF A SET

In mathematics, a **partition** of a set refers to a division of its elements into non-empty, mutually exclusive subsets, such that each element of the original set belongs to exactly one of these subsets. In simpler terms, a partition is a "set of sets", where each subset is known as a **cell**.

**Definition 5 (Partition of A Set** (Brualdi, 2004)**).** *A family of sets $\mathcal{P}$ is a partition of the set $X$ if and only if all of the following conditions hold:*

*(1) $\mathcal{P}$ does not contain the empty set (i.e., $\emptyset \notin P$).*

*(2) The union of the sets in $\mathcal{P}$ is equal to $X$ (i.e., $\bigcup_{A \in \mathcal{P}} A = X$). The sets in $\mathcal{P}$ are said to exhaust or cover $X$.*

*(3) The intersection of any two distinct sets in $\mathcal{P}$ is empty (i.e., $\forall A, B \in \mathcal{P}, A \neq B \implies A \cap B = \emptyset$). The elements of $\mathcal{P}$ are said to be pairwise disjoint or mutually exclusive.*

In this work, we introduce two specialized types of partitions applied to the latent variable set $\mathcal{Z}$: the **single-view partition** $\mathcal{P}_s(\mathcal{Z})$ and the **complete-view partition** $\mathcal{P}_c(\mathcal{Z})$. These partitions are tailored to the particular structure and requirements of the problem under study.

The visualization of these two partitions is facilitated by arranging the variables in a matrix and dividing them according to rows and columns. In Figure 5, the left matrix $Z_0$ contains diagonal elements directly derived from observed data, while off-diagonal elements represent transformations of the diagonal elements. Each column consists of variables $z_*^{(l)}$ corresponding to the $l$-th view, derived from different sources. Hence, the single-view partition of $\mathcal{Z}$ corresponds to the set of columns in the matrix. In contrast, the complete-view partition can be represented by dividing the matrix by rows, where each row encompasses all $L$ views. However, this partition is not unique; by reordering the elements within each column and then partitioning the matrix by rows, we obtain a new complete-view partition $\mathcal{P}'_c(\mathcal{Z})$, as illustrated by the right matrix $Z_1$ in Figure 5.

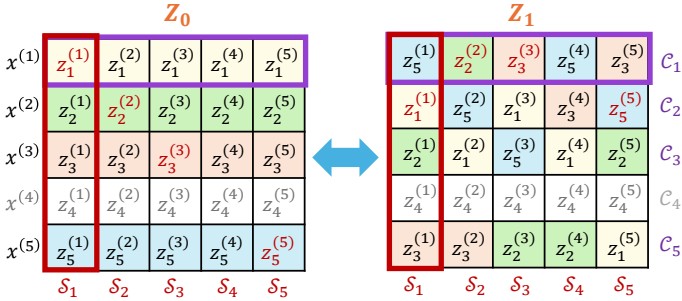

Figure 5: An illustration of column-wise permutations to generate different complete-view partitions. In the first column (red box), the permutation $\sigma_1 = (1532)(4)$ is applied, and its inverse $(\sigma_1)^{-1} = (2351)(4)$ reverses the cycle order. This results in $\sigma_1([z_1^{(1)}, z_2^{(1)}, z_3^{(1)}, z_4^{(1)}, z_5^{(1)}]) = [z_5^{(1)}, z_1^{(1)}, z_2^{(1)}, z_4^{(1)}, z_3^{(1)}]$. The same procedure is applied to the other columns. Partitioning each row (purple box) yields the complete-view partition.

Next, we define **permutations**, which are central to generating new partitions.

**Definition 6 (Permutation of A Set** (Bóna, 2008)**).** *A permutation of a set $X$ is a bijective function $\sigma : X \to X$. In other words, it is a one-to-one mapping of the set $X$ onto itself.*

A common method for representing permutations is **cycle notation**, where a permutation is expressed as a product of disjoint cycles. Each cycle indicates how the permutation rearranges a

subset of elements, moving each element to the position of the next one in the cycle. For example, consider a permutation $\sigma$ of the set $X = 1, 2, 3, 4$, defined by $\sigma(1) = 2$, $\sigma(2) = 3$, $\sigma(3) = 1$, and $\sigma(4) = 4$. In cycle notation, this permutation is written as $\sigma = (123)(4)$. The cycle $(123)$ indicates that 1 is mapped to 2, 2 to 3, and 3 back to 1. The element 4 remains fixed, represented as $(4)$, often referred to as a **fixed point**.

### A.2 CYCLIC PERMUTATION AND SATTOLO'S ALGORITHM

A cyclic permutation is a specific type of permutation that consists of exactly one cycle in its cycle notation, with the cycle length equal to the size of the set (Gross, 2016). For example, a cyclic permutation $\sigma$ of the set $X = \{1, 2, 3, 4\}$ can be written as $(k_1 k_2 k_3 k_4)$, where $k_i \in X$. A formal definition is given in Definition 3. In a cyclic permutation, each element of a set with more than one element is cyclically shifted, meaning each element is mapped to another, and after a number of mappings equal to the set size, every element returns to its original position. Cyclic permutations are particularly useful in our setting, as they guarantee convergence of the regularization term (see Appendix A.3) and can be efficiently generated using **Sattolo's Algorithm** (Wilson, 2005), which operates with linear time complexity.

---

**Algorithm 1** Sattolo's Algorithm for Cyclic Permutation

**Input:** Array $A$ of size $n$
**Output:** A cyclic permutation of $A$
Initialize array length $n = |A|$.
  1: **for** $i = n - 1$ to 1 **do**
  2:     Randomly select $j$ from 0 to $i - 1$;
  3:     Swap $A[i]$ and $A[j]$;
  4: **end for**
  5: **return** $A$;

---

The algorithm starts with the identity permutation $\sigma^{(0)} = Id$. For each $i \in \{1, \ldots, n-1\}$, we denote by $\sigma^{(i)}$ the permutation obtained after the $i$ first steps. Step $i$ consists in choosing a random integer $k_i$ in $\{1, \ldots, n-i\}$ and swapping the values of $\sigma^{(i-1)}$ at places $k_i$ and $n-i+1$. In this way, we obtain a new permutation $\sigma^{(i)}$, which is equal to $\tau_{k_i, n-i+1} \circ \sigma^{(i-1)}$, where $\tau_{k_i, n-i+1}$ is the transposition exchanging $k_i$ and $n - i + 1$. Finally, the algorithm returns the permutation $\sigma = \sigma^{(n-1)}$. This process is captured in Algorithm 1. Figure 6 provides an example where $n = 5$, and the sequence of random swaps is $3, 1, 2, 1$. The resulting cyclic permutation is $1 \rightarrow 5 \rightarrow 3 \rightarrow 2 \rightarrow 4 \rightarrow 1$.

$$
\begin{array}{ccccc}
1 & 2 & \underline{3} & 4 & \mathbf{5} \\
\underline{1} & 2 & 5 & \mathbf{4} & 3 \\
4 & \underline{2} & \mathbf{5} & 1 & 3 \\
\underline{4} & \mathbf{5} & 2 & 1 & 3 \\
5 & 4 & 2 & 1 & 3 \\
\end{array}
$$

Figure 6: Execution of Sattolo's algorithm for $n = 5$, where the sequence of random swaps is $3, 1, 2, 1$ (Wilson, 2005).

**Proposition 1.** *The mapping produced by Sattolo's algorithm is a cyclic permutation, and every cyclic permutation can be obtained using Sattolo's algorithm.*

**Proof.** The correctness of Sattolo's algorithm follows from the fact that it generates a unique decomposition of a cyclic permutation $\sigma$ as a product of transpositions, $\tau_{k_{n-1}, 2} \circ \cdots \circ \tau_{k_i, n-i+1} \circ \cdots \circ \tau_{k_1, n}$, where $k_i \in \{1, \ldots, n-i\}$ for $1 \leq i \leq n-1$.

For $n = 2$, the permutation $\sigma = \tau_{k_1, 2}$ is clearly a cyclic permutation. Assuming Sattolo's algorithm works for sets with fewer than $n$ elements, we now demonstrate that it also holds for a set of size $n$.

Consider a set of $n$ elements. The permutation $\sigma = \tau_{k_{n-1},2} \circ \cdots \circ \tau_{k_i,n-i+1} \circ \cdots \circ \tau_{k_2,n-1} \circ \tau_{k_1,n}$ consists of $n-1$ transpositions. The first $n-2$ transpositions act on the set $\{1,\ldots,n-1\}$, excluding $k_1$, where $k_1$ is swapped with $n$. By the inductive hypothesis, these form a cyclic permutation on $n-1$ elements, which can be represented as a single cycle: $\tau_{k_{n-1},2} \circ \cdots \circ \tau_{k_i,n-i+1} \circ \cdots \circ \tau_{k_2,n-1} = (n, r_1, \cdots r_{n-2})$.

Applying the final transposition $\tau_{k_1,n}$ swaps $k_1$ with $n$, yielding:

$$\sigma = (n, r_1, \cdots r_{n-2}) \circ \tau_{k_1,n} = (n, k_1, r_1, \cdots r_{n-2}),$$

which is a cyclic permutation of $n$ elements. By induction, Sattolo's algorithm always produces a cyclic permutation for any set size $n$.

Since Sattolo's algorithm generates $(n-1)!$ distinct cyclic permutations for a set of size $n$, and this is precisely the number of all possible cyclic permutations on $n$ elements, every cyclic permutation can be obtained using this method. This completes the proof. $\square$

Next, we explain how permutations generate different **complete-view partitions**. Let $v$ represent rows and $l$ columns, with the matrix form of the set $\boldsymbol{\mathcal{Z}}$ expressed as $Z_0 = \left[z_v^{(l)}\right]_{L \times L}$. We assume the missing views preserve the matrix's square form, simplifying notation. Let the observed views be $\mathcal{I} = \{k_1, \ldots, k_n\} \subseteq [L]$, with missing views given by $[L] \setminus \mathcal{I} = \{r_1, \ldots, r_m\}$. A complete-view partition is obtained by dividing $Z_0$ by rows: $\mathcal{P}_c(\boldsymbol{\mathcal{Z}}; \text{Id}) = \{\boldsymbol{\mathcal{C}}_n^0\}_{n \in \mathcal{I}}$, where $\boldsymbol{\mathcal{C}}_n^0 = \{z_n^{(l)}\}_{l=1}^L$, with Id representing the identity mapping. We call $\mathcal{P}_c(\boldsymbol{\mathcal{Z}}; \text{Id})$ the **basic complete-view partition**.

The matrix $Z_0$ can also be expressed by columns as $\left[z_v^{(l)}\right]_{L \times L} = \left[\boldsymbol{z}^{(1)}, \cdots, \boldsymbol{z}^{(L)}\right]$. Consider $L$ permutations $\boldsymbol{\sigma} = \{\sigma_l\}_{l=1}^L$ defined on the index set $[L]$, each being a cyclic permutation on $\{k_1, \cdots, k_n\}$ with $m$ fixed points $\{r_1, \cdots, r_m\}$. A new matrix $Z_1$ can then be constructed by permuting the row indices within each column $\left[z_{\sigma_l(v)}^{(l)}\right]_{L \times L} = \left[\tilde{\boldsymbol{z}}^{(1)}, \cdots, \tilde{\boldsymbol{z}}^{(L)}\right]$, where $\tilde{\boldsymbol{z}}^{(l)}$ contains the same elements as $\boldsymbol{z}^{(l)}$, which implies that single-view partition is unique. And this results in a new $\mathcal{P}_c(\boldsymbol{\mathcal{Z}}; \boldsymbol{\sigma}) = \{\boldsymbol{\mathcal{C}}_n\}_{n \in \mathcal{I}}$, where $\boldsymbol{\mathcal{C}}_n = \{z_{\sigma_l(n)}^{(l)}\}_{l=1}^L$.

As illustrated in Figure 5, if we disregard the missing views treated as fixed points, applying a cyclic permutation transforms matrix $Z_0$ into $Z_1$. Notably, applying the inverse of the permutation, which is also cyclic, restores $Z_1$ back to $Z_0$. This highlights a symmetric relationship between the two matrices, governed by cyclic permutations and their inverses.

**Proposition 2.** *The inverse of a cyclic permutation is also a cyclic permutation.*

**Proof.** We prove this using a convenient feature of the cycle notation. Consider a cyclic permutation $\sigma$ defined on the set $[L]$, with cycle notation $(k_1, k_2, \cdots, k_L)$. The inverse permutation $\sigma^{-1}$ is obtained by reversing the order of the elements in the cycle, yielding $(k_L, k_{L-1}, \cdots, k_1)$. Since this reversed sequence still forms a single cycle, $\sigma^{-1}$ is indeed a cyclic permutation. $\square$

## A.3 THE SIMILARITY MEASURE OVER DISTRIBUTIONS

In this section, we introduce a similarity measure between distributions, referred to as the **Dissimilarity Coefficient** ($d.c.$), which we use to reduce the value of the KL divergence term in the ELBO. Consequently, this reduction minimizes the dissimilarity between distributions, effectively maximizing their similarity. We also explain how the informational prior in our method transforms the regularization term into a new $d.c.$.

In various statistical fields—such as hypothesis testing, cluster analysis, and pattern recognition—it is essential to distinguish between probability distributions using appropriate dissimilarity coefficients (or separation measures, denoted as $d.c.$). A $d.c.$ for a set of $N$ probability distributions quantifies their "degree of heterogeneity". For $N = 2$, a $d.c.$ can be interpreted as a "distance" between two distributions, though it may not always represent a metric distance in the strict sense.

**Definition 7 (Dissimilarity Coefficient** (Sgarro, 1981)**).** *Let $\mathcal{P}$ denote the set of probability measures on a measurable space $(\Omega, \mathcal{F})$. Let $N \geq 2$ be a fixed natural number. A dissimilarity coefficient ($d.c.$) of order $N$ is a mapping $d : \mathcal{P}^N \to \mathbb{R} \cup \{+\infty\}$ that satisfies the following properties for any $P_1, P_2, \ldots, P_N \in \mathcal{P}$:*

(1) *Non-negativity:* $d\left(P_1, P_2, \ldots, P_N\right) \geq 0$;

(2) *Identity of Indiscernible:* $d\left(P_1, P_2, \ldots, P_N\right) = 0$ *if* $P_1 = P_2 = \cdots = P_N$;

(3) *Symmetry:* $d\left(P_1, P_2, \ldots, P_N\right) = d\left(P_{\phi(1)}, P_{\phi(2)}, \ldots, P_{\phi(N)}\right)$ *for any permutation* $\phi$ *of* $[N]$.

*In some cases, condition (2) can be strengthened to:*

(2') $d\left(P_1, P_2, \ldots, P_N\right) = 0$ *if and only if* $P_1 = P_2 = \cdots = P_N$.

The Kullback-Leibler (KL) divergence (Kullback, 1959), while not symmetric, is widely regarded as a fundamental statistical measure for distinguishing between two probability distributions. To extend its application to multiple distributions ($N \geq 2$), several symmetric dissimilarity coefficients based on the KL divergence have been proposed. For example, the average divergence sums the KL divergence over all $N(N-1)$ pairs of distributions (Kullback, 1959), while the information radius resembles the Jensen-Shannon divergence for multiple distributions (Sibson, 1969; Jardine & Sibson, 1971). Additionally, Sgarro (1981) introduced the minimum divergence, which measures the KL divergence for the pair with the smallest difference.

In this work, we reuse the permuted posteriors generated during the construction of complete-view partitions and set them as priors within the multimodal VAE framework. This transforms the KL divergence term in the ELBO into a new $d.c.$, which we call the **Permutation Divergence** (Definition 4). We prove that this is a valid $d.c.$:

**Theorem 1.** *The Permutation Divergence defined in Definition 4 is a dissimilarity coefficient.*

**Proof.** Consider a cyclic permutation $\sigma$ defined on $[N]$ with cycle notation $(1, r_1, \ldots, r_{N-1})$. The corresponding permutation divergence is given by:

$$d(P_1, P_2, \ldots, P_N; \sigma) = \sum_{i=1}^{N} \mathrm{KL}[P_i \parallel P_{\sigma(i)}].$$

Since the divergence is a sum of KL divergences, property (1) of non-negativity is satisfied.

To verify the identity of indiscernible, note that $d(P_1, P_2, \ldots, P_N; \sigma) = 0$ if and only if each term in the sum is zero. This implies $\mathrm{KL}[P_i \parallel P_{\sigma(i)}] = 0$ for all $i$, which occurs if and only if $P_i = P_{\sigma(i)}$. Consequently, $P_1 = P_{r_1} = \cdots = P_{r_{N-1}}$, satisfying property (2').

Finally, for any permutation $\phi$ of $[N]$:

$$d\left(P_{\phi(1)}, P_{\phi(2)}, \ldots, P_{\phi(N)}; \sigma\right) = \sum_{i=1}^{N} \mathrm{KL}[P_{\phi(i)} \parallel P_{\sigma(\phi(i))}]$$

$$= \sum_{j=1}^{N} \mathrm{KL}[P_j \parallel P_{\sigma(j)}] = d(P_1, P_2, \ldots, P_N; \sigma).$$

Thus, the Permutation Divergence satisfies the symmetry property (3). □

The proof of property (2') demonstrates why cyclic permutations are used instead of general permutations: the one-cycle structure ensures that the divergence reaches its minimum when all distributions are identical.

**Proposition 3.** *The sum of dissimilarity coefficients defined on the same set of distributions is itself a dissimilarity coefficient.*

**Proof.** The proofs of these properties for the sum of $d.c.$'s follow directly from the corresponding properties of the individual coefficients. For brevity, these straightforward proofs are omitted here. □

As a result, the sum of two Permutation Divergences, each defined with different cyclic permutations on the same set of distributions, remains a valid $d.c.$. Importantly, this applies to the case where the cyclic permutation and its inverse are used together.

**Proposition 4.** *The sum of the Permutation Divergences defined with a cyclic permutation $\sigma$ and its inverse $\sigma^{-1}$ is also a dissimilarity coefficient and is composed of symmetric KL divergences.*

**Proof.** Consider a cyclic permutation $\sigma$ defined on $[N]$. The corresponding permutation divergence is given by:

$$d(P_1, P_2, \ldots, P_N; \sigma) = \sum_{i=1}^{N} \text{KL}[P_i \parallel P_{\sigma(i)}].$$

According to Proposition 3, $\sigma^{-1}$ is also a cyclic permutation, thus

$$d(P_1, P_2, \ldots, P_N; \sigma^{-1}) = \sum_{j=1}^{N} \text{KL}[P_j \parallel P_{\sigma^{-1}(j)}].$$

For any $j \in [N]$, there exists a unique $k \in [N]$ such that $\sigma(k) = j$. Therefore, we have:

$$\sum_{i=1}^{N} \text{KL}[P_i \parallel P_{\sigma(i)}] + \sum_{j=1}^{N} \text{KL}[P_j \parallel P_{\sigma^{-1}(j)}]$$
$$= \sum_{i=1}^{N} \text{KL}[P_i \parallel P_{\sigma(i)}] + \sum_{k=1}^{N} \text{KL}[P_{\sigma(k)} \parallel P_{\sigma^{-1}(\sigma(k))}]$$
$$= \sum_{i=1}^{N} \left( \text{KL}[P_i \parallel P_{\sigma(i)}] + \text{KL}[P_{\sigma(i)} \parallel P_i] \right)$$
$$\triangleq d(P_1, P_2, \ldots, P_N; \sigma, \sigma^{-1}),$$

where each term is symmetric with respect to the pair $P_i$ and $P_{\sigma(i)}$. $\qquad\qquad\square$

As illustrated in Figure 5, a cyclic permutation transforms one complete-view partition into a new one, while its inverse restores the original partition. This allows for the interchangeability of the matrices, $Z_0$ and $Z_1$, where one can be used for posterior factorization and the other for priors. Consequently, examining the sum of the Permutation Divergences defined by a cyclic permutation and its inverse highlights this symmetry, which will be further explored in the next section.

## A.4 Derivation and Analysis of the Evidence Lower Bound (ELBO)

With the complete-view partition factorizing the joint approximate posterior and the permuted unimodal posteriors serving as informational priors, we are now ready to derive the ELBO for incomplete multi-view data $\{x^{(v)}\}_{\mathcal{I}}$. To facilitate this derivation, which involves two types of latent variables, we first present a useful lemma that establishes the chain rule for KL divergence.

**Lemma 1** (Chain Rule of KL divergence).
$$KL(q(x, y) \| p(x, y)) = KL(q(x) \| p(x)) + KL(q(y|x) \| p(y|x))$$

**Proof.** The proof follows from the definition of the KL divergence and the factorization of joint distributions:

$$\text{KL}(q(x, y) \| p(x, y)) = \int \int q(x, y) \log \frac{q(x, y)}{p(x, y)} \, dy \, dx$$
$$= \int \int q(x, y) \log \frac{q(x) q(y|x)}{p(x) p(y|x)} \, dy \, dx$$
$$= \int \int q(x, y) \log \frac{q(x)}{p(x)} \, dy \, dx + \int \int q(x, y) \log \frac{q(y|x)}{p(y|x)} \, dy \, dx$$
$$= \int q(x) \log \frac{q(x)}{p(x)} \, dx + \int q(x) \int q(y|x) \log \frac{q(y|x)}{p(y|x)} \, dy \, dx$$
$$= \text{KL}(q(x) \| p(x)) + \text{KL}(q(y|x) \| p(y|x)).$$

$\qquad\qquad\square$

**Theorem 2.** *Equation ([1](#)) is the evidence lower bound (ELBO) for incomplete multi-view data $\{x^{(v)}\}_{\mathcal{I}}$.*

**Proof.** We begin with the log-likelihood of the incomplete multi-view data $\{x^{(v)}\}_{\mathcal{I}}$, assuming two sets of latent variables, $\mathbf{Z}$ and $\mathbf{\Omega}$. For any joint distribution $q(\mathbf{Z}, \mathbf{\Omega})$, the following equation holds:

$$\log p(\{x^{(v)}\}_{\mathcal{I}}) = \int q(\mathbf{Z}, \mathbf{\Omega}) \log p(\{x^{(v)}\}_{\mathcal{I}}) d\mathbf{Z} d\mathbf{\Omega}$$

$$= \int q(\mathbf{Z}, \mathbf{\Omega}) \log \frac{p(\{x^{(v)}\}_{\mathcal{I}} \mid \mathbf{Z}, \mathbf{\Omega}) p(\mathbf{Z}, \mathbf{\Omega})}{p(\mathbf{Z}, \mathbf{\Omega} \mid \{x^{(v)}\}_{\mathcal{I}})} \frac{q(\mathbf{Z}, \mathbf{\Omega})}{q(\mathbf{Z}, \mathbf{\Omega})} d\mathbf{Z} d\mathbf{\Omega}$$

$$= \underbrace{\int q(\mathbf{Z}, \mathbf{\Omega}) \log \frac{p(\{x^{(v)}\}_{\mathcal{I}}, \mathbf{Z}, \mathbf{\Omega})}{q(\mathbf{Z}, \mathbf{\Omega}))} d\mathbf{Z} d\mathbf{\Omega}}_{\text{Evidence Lower Bound (ELBO)}} + \text{KL}(q(\mathbf{Z}, \mathbf{\Omega}) \| \underbrace{p(\mathbf{Z}, \mathbf{\Omega} \mid \{x^{(v)}\}_{\mathcal{I}})}_{\text{True posterior}})$$

We use encoders to model the distribution $q(\mathbf{Z}, \mathbf{\Omega})$ given the observed data, denoted as $q(\mathbf{Z}, \mathbf{\Omega} \mid \{x^{(v)}\}_{\mathcal{I}})$. The first term in this equation represents the ELBO, which serves as a lower bound on the log-likelihood of the data. By maximizing the ELBO, the KL term becomes smaller, meaning that the learned distribution approximates the true posterior $p(\mathbf{Z}, \mathbf{\Omega} \mid \{x^{(v)}\}_{\mathcal{I}})$. For a given complete-view partition $\mathcal{P}_c(\mathbf{Z}; \boldsymbol{\sigma}) = \{\mathcal{C}_n\}_{n \in \mathcal{I}}$, where $\mathcal{C}_n = \{z_{\sigma_l(n)}^{(l)}\}_{l=1}^L$, we can factorize the joint posterior as:

$$q(\mathbf{Z}, \mathbf{\Omega} \mid \{\boldsymbol{x}^{(v)}\}_{\mathcal{I}}) \triangleq \prod_{n \in \mathcal{I}} q(\boldsymbol{\omega}_n \mid \mathcal{C}_n, \{\boldsymbol{x}^{(v)}\}_{\mathcal{J}_n}) q(\mathcal{C}_n \mid \{\boldsymbol{x}^{(v)}\}_{\mathcal{J}_n})$$

$$= \prod_{n \in \mathcal{I}} q(\boldsymbol{\omega}_n \mid \mathcal{C}_n, \{\boldsymbol{x}^{(v)}\}_{\mathcal{J}_n}) \prod_{l=1, v \in \mathcal{J}_n}^L q(\boldsymbol{z}_v^{(l)} \mid \boldsymbol{x}^{(v)}).$$

Next, we assume the generative process as:

$$p(\{x^{(v)}\}_{\mathcal{I}}, \mathbf{Z}, \mathbf{\Omega}) \triangleq \prod_{n \in \mathcal{I}} p(x^{(n)} | \mathcal{C}_n, \omega_n) p(\mathcal{C}_n, \omega_n)$$

$$= \prod_{n \in \mathcal{I}} p\left(x^{(n)} | \mathcal{C}_n \cap \boldsymbol{S}_n, \omega_n\right) p(\omega_n) \prod_{l=1, v \in \mathcal{J}_n}^L p\left(z_v^{(l)}\right).$$

Here, we again explain why we use $\mathcal{C}_n \cap \boldsymbol{S}_n$, the diagonal of the matrix of $\mathbf{Z}$, for reconstruction. From the derivation, we partition according to $\{\mathcal{C}_n\}_{n \in \mathcal{I}}$, where each $\mathcal{C}_n$ contains $L$ latent variables representing different views. Among them, only $z_{\sigma_l(n)}^{(n)} = \mathcal{C}_n \cap \boldsymbol{S}_n$ is related to $x^{(n)}$ (with the same superscript). This approach also simplifies practical implementation, as we can directly use the diagonal of the matrix of $\mathbf{Z}$ to extract all the $\mathcal{C}_n \cap \boldsymbol{S}_n$.

$$\int q(\mathbf{Z}, \mathbf{\Omega} \mid \{x^{(v)}\}_{\mathcal{I}}) \log \frac{p(\{x^{(v)}\}_{\mathcal{I}}, \mathbf{Z}, \mathbf{\Omega})}{q(\mathbf{Z}, \mathbf{\Omega} \mid \{x^{(v)}\}_{\mathcal{I}})} d\mathbf{Z} d\mathbf{\Omega}$$

$$= \sum_{n \in \mathcal{I}} \left\{ \int q(\mathcal{C}_n, \omega_n \mid \{x^{(v)}\}_{\mathcal{J}_n}) \log \frac{p(x^{(n)} \mid \mathcal{C}_n, \omega_n) p(\omega_n) \prod_{l=1, v \in \mathcal{J}_n}^L p(z_v^{(l)})}{q(\omega_n \mid \mathcal{C}_n, \{x^{(v)}\}_{\mathcal{J}_n}) \prod_{l=1, v \in \mathcal{J}_n}^L q(z_v^{(l)} \mid x^{(v)})} d\mathcal{C}_n d\omega_n \right\}$$

$$= \sum_{n \in \mathcal{I}} \left\{ \mathbb{E}_{q(\mathcal{C}_n \cap \boldsymbol{S}_n, \omega_n \mid \{x^{(v)}\}_{\mathcal{J}_n})} \left[ \log p(x^{(n)} \mid \mathcal{C}_n \cap \boldsymbol{S}_n, \omega_n) \right] \right.$$

$$+ \int q(\omega_n \mid \mathcal{C}_n, \{x^{(v)}\}_{\mathcal{J}_n}) \log \frac{p(\omega_n)}{q(\omega_n \mid \mathcal{C}_n, \{x^{(v)}\}_{\mathcal{J}_n})} d\omega_n$$

$$+ \left. \int \prod_{l=1, v \in \mathcal{J}_n}^L q(z_v^{(l)} \mid x^{(v)}) \log \frac{\prod_{l=1, v \in \mathcal{J}_n}^L p(z_v^{(l)})}{\prod_{l=1, v \in \mathcal{J}_n}^L q(z_v^{(l)} \mid x^{(v)})} d\mathcal{C}_n \right\}$$

$$= \sum_{n \in \mathcal{I}} \mathbb{E}_{q(\mathcal{C}_n, \omega_n \mid \{x^{(v)}\}_{\mathcal{J}_n})} \left[ \log p(x^{(n)} \mid \mathcal{C}_n \cap \boldsymbol{S}_n, \omega_n) \right]$$

$$- \sum_{l=1}^L \sum_{v \in \mathcal{I}} KL \left[ q(z_v^{(l)} \mid x^{(v)}) \| p(z_v^{(l)}) \right] - \sum_{n \in \mathcal{I}} KL \left( q(\omega_n \mid \mathcal{C}_n, \{x^{(v)}\}_{\mathcal{I}_n}) \| p(\omega_n) \right).$$

The split of the final two KL terms follows directly from the chain rule provided in the lemma. At this point, we have derived Eq. (1), but we can further simplify by removing redundant variables (which are essential for modeling and derivation but not necessary for exposition) and explicitly defining the prior settings, as outlined in Section 3.3. Specifically, we denote $q(z_v^{(l)} \mid x^{(v)})$, which encodes the $l$-th view's information using the $v$-th view as the source, as $q_v^{(l)}(z)$. his notation represents the encoding and transformation of the distribution, where $q_v^{(l)}(z) \sim \mathcal{N}(z; f_{lv} \circ \mu(x^{(v)}), f_{lv} \circ \Sigma(x^{(v)}))$, where $f_{lv} = \mathrm{id}$ when $l = v$. Similarly, we denote the prior $p(z_v^{(l)})$ as $p_v^{(l)}(z)$. We set this prior to $p_v^{(l)}(z) = q_{\sigma_l^{-1}(v)}^{(l)}(z)$, where $\sigma_l^{-1}$ is the inverse of the permutation used to obtain the complete-view partition and acts as a cyclic permutation on the incomplete index set $\mathcal{I}$. This transforms the first KL term into:

$$\sum_{l=1}^{L} \sum_{v \in \mathcal{I}} KL\left[q_v^{(l)}(z) \parallel q_{\sigma_l^{-1}(v)}^{(l)}(z)\right] = \sum_{l=1}^{L} d(q_{k_1}^{(l)}, \ldots, q_{k_n}^{(l)}; \sigma_l^{-1}),$$

where $\{k_1, \ldots, k_n\}$ represents the observed views, and $d$ is the permutation divergence, ensuring that distributions encoding the same view from different sources remain as close as possible. For practical implementation, we only need to compute the KL divergence between the corresponding positions of the distributions in the left and right matrices shown in Figure 5.

The latent variable $\omega$ is derived from the fusion of the marginal Gaussian distributions in $\mathcal{C}_n$. In other words, $\omega$ can be deterministically determined by the variable $z$, making the regularization of $\omega$ effectively a regularization of $z$. We simplify the notation for the posterior distribution $q(\omega \mid \mathcal{C}_n)$, which represents the distribution obtained by fusing the marginal $k$-dimensional distributions. Specifically, $q(\omega \mid \mathcal{C}_n) \sim \mathcal{N}(\omega; \alpha_n, \Lambda_n)$, where

$$\Lambda_n = \left[\sum_{l=1}^{L} \Sigma_v^{(l)}(1:k, 1:k)^{-1}\right]^{-1}, \alpha_n = \Lambda_n \sum_{l=1}^{L} \left[\Sigma_v^{(l)}(1:k, 1:k)^{-1} \mu_v^{(l)}(1:k)\right], v \in \mathcal{J}_n.$$

Here, we rely on two well-known results: first, the marginal distribution of a multivariate Gaussian remains Gaussian, and second, the geometric mean of several Gaussian random variables also follows a Gaussian distribution, with parameters that are straightforward to compute.

For the prior setting of $\omega$, since its posterior is derived from the combination $\mathcal{C}_n = \{z_{\sigma_l(n)}^{(l)}\}_{l=1}^{L}$, we can similarly fuse the priors of $z$, which have already been defined. It is easy to see that $\{z_{\sigma_l^{-1}(\sigma_l(n))}^{(l)})^{(l)}\}_{l=1}^{L} = \{z_n^{(l)}\}_{l=1}^{L} = \mathcal{C}_n^0$, representing the cell of the basic complete-view partition, which corresponds to the pre-permutation position in the matrix. Therefore, we set the prior of $q(\omega \mid \mathcal{C}_n)$ to $q(\omega \mid \mathcal{C}_n^0)$. In practice, this simply requires calculating the KL divergence between the $\omega$ variables obtained from the two matrices shown in Figure 5.

Finally, for given permutations and their resulting complete-view partition, we can express the ELBO as follows:

$$\mathcal{L}_{\mathrm{ELBO}}(\{\boldsymbol{x}^{(v)}\}_{\mathcal{I}}) = \sum_{n \in \mathcal{I}} \mathbb{E}_{q(\mathcal{C}_n, \omega \mid \{x^{(v)}\}_{\mathcal{J}_n})} \left[\log p(x^{(n)} \mid z_{\sigma_l(n)}^{(n)}, \omega)\right]$$

$$- \sum_{l=1}^{L} \sum_{v \in \mathcal{I}} KL\left[q_v^{(l)}(z) \parallel q_{\sigma_l^{-1}(v)}^{(l)}(z)\right] - \sum_{n \in \mathcal{I}} KL\left(q(\omega \mid \mathcal{C}_n) \parallel q(\omega \mid \mathcal{C}_n^0)\right). \tag{2}$$

$\square$

Since we can factorize the joint posterior using any complete-view partition, we can alternatively use the basic partition $\mathcal{P}_c^0(\boldsymbol{\mathcal{Z}}) = \{\mathcal{C}_n^0\}_{n \in \mathcal{I}}$, where $\mathcal{C}_n^0 = \{z_n^{(l)}\}_{l=1}^{L}$. In this case, there are no cyclic permuted posteriors to set the prior. Thus, assuming arbitrary cyclic permutations $\boldsymbol{\sigma} = \{\sigma_l\}_{l=1}^{L}$ on the incomplete index set $\mathcal{I}$, we can derive the basic ELBO as follows:

$$\mathcal{L}_{\mathrm{ELBO}}^0(\{\boldsymbol{x}^{(v)}\}_{\mathcal{I}}) = \sum_{n \in \mathcal{I}} \mathbb{E}_{q(\mathcal{C}_n^0, \omega \mid x^{(n)})} \left[\log p(x^{(n)} \mid z_n^{(n)}, \omega)\right]$$

$$- \sum_{l=1}^{L} \sum_{v \in \mathcal{I}} KL\left[q_v^{(l)}(z) \parallel q_{\sigma_l(v)}^{(l)}(z)\right] - \sum_{n \in \mathcal{I}} KL\left(q(\omega \mid \mathcal{C}_n^0) \parallel q(\omega \mid \mathcal{C}_n)\right). \tag{3}$$

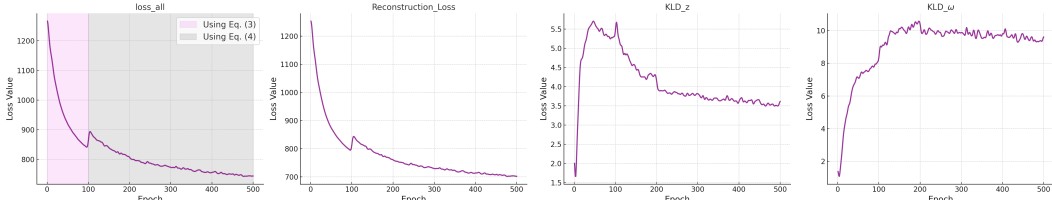

Figure 7: Loss evolution curves during the training process on the Handwritten under missing rate $\eta = 0.1$, illustrating the trends of different loss components. The first subplot (Loss all) highlights the warm-up phase using Eq. (3) during the initial 100 epochs (shaded yellow). Afterward, training transitions to Eq. (4), which is defined as $0.5 \times$ (Eq. (2) + Eq. (3)). Subplots for Reconstruction Loss, KLD_z, and KLD_$\omega$ show the progressive change in their respective values over 500 epochs.

There are subtle differences between the basic ELBO in Eq.(3) and Eq.(2). The first term, representing the reconstruction loss, shows that in the basic ELBO, the variable $z_n^{(n)}$ generates $x^{(n)}$ through **self-view reconstruction**, meaning $z_n^{(n)}$ is directly encoded from $x^{(n)}$ without any transformations. In contrast, in Eq.(2), $z_{\sigma_l(n)}^{(n)}$ comes from a cyclically permuted complete-view partition, where $\sigma_l(n)$ is not equal to $n$ but instead corresponds to another observed view obtained via inter-view correspondences. This can be interpreted as a **cross-view generation**. If Eq.(2) iterates over all possible permutations, it implies that within the loss function, all observed views generate other views via cross-generation.

The form of the remaining two terms suggests that we can form a convex combination of these two types of ELBOs, assuming they use the same set of $\boldsymbol{\sigma}$ and their inverses. This leads to the following expression:

$$
\begin{aligned}
\mathcal{L}_{\text{ELBO}}(\{\boldsymbol{x}^{(v)}\}_{\mathcal{I}}; \boldsymbol{\sigma}) &= \frac{1}{2}(\mathcal{L}_{\text{ELBO}}^0(\{\boldsymbol{x}^{(v)}\}_{\mathcal{I}}) + \mathcal{L}_{\text{ELBO}}(\{\boldsymbol{x}^{(v)}\}_{\mathcal{I}})) \\
&= \frac{1}{2}\sum_{n \in \mathcal{I}} \left\{ \mathbb{E}_{q(\mathcal{C}_n^0, \omega | x^{(n)})} \left[ \log p(x^{(n)} \mid z_n^{(n)}, \omega_n) \right] + \mathbb{E}_{q(\mathcal{C}_n, \omega | \{x^{(v)}\}_{\mathcal{J}_n})} \left[ \log p(x^{(n)} \mid z_{\sigma_l(n)}^{(n)}, \omega) \right] \right\} \\
&\qquad - \frac{1}{2}\sum_{l=1}^{L}\sum_{v \in \mathcal{I}} \left\{ KL\left[ q_v^{(l)}(z) \parallel q_{\sigma_l^{-1}(v)}^{(l)}(z) \right] + KL\left[ q_v^{(l)}(z) \parallel q_{\sigma_l(v)}^{(l)}(z) \right] \right\} \\
&\qquad - \frac{1}{2}\sum_{n \in \mathcal{I}} \left\{ KL\left( q(\omega \mid \mathcal{C}_n) \parallel q(\omega \mid \mathcal{C}_n^0) \right) + KL\left( q(\omega \mid \mathcal{C}_n^0) \parallel q(\omega \mid \mathcal{C}_n) \right) \right\}
\end{aligned}
\tag{4}
$$

In practical optimization, we use Eq.(3) for warm up and Eq.(4) as the final learning objective, selecting a different $\boldsymbol{\sigma}$ permutation at each iteration. This approach offers three main advantages. **First**, combining the two ELBOs simultaneously promotes both self-view reconstruction and cross-view generation, which leads to a more comprehensive learning process. **Second**, by using cyclic permutations $\boldsymbol{\sigma}$ and exploiting the fact that their inverses are also cyclic, we can efficiently obtain two complete-view partitions. This allows the distributions at corresponding positions, both pre- and post-permutation, to supervise each other without the need for additional computations, making the optimization process more computationally efficient. **Finally**, the convex combination of the two regularization terms introduces a higher degree of symmetry. For the term involving $z$, based on Proposition 4, we can rewrite it as a more symmetric dissimilarity coefficient:

$$
\sum_{l=1}^{L}\sum_{v \in \mathcal{I}} \left\{ KL\left[ q_v^{(l)}(z) \parallel q_{\sigma_l^{-1}(v)}^{(l)}(z) \right] + KL\left[ q_v^{(l)}(z) \parallel q_{\sigma_l(v)}^{(l)}(z) \right] \right\} = \sum_{l=1}^{L} d(q_{k_1}^{(l)}, \ldots, q_{k_n}^{(l)}; \sigma_l, \sigma_l^{-1}).
$$

Similarly, the term involving $\omega$ is also expressed as a sum of symmetric KL divergences. In prior work, such as Sutter et al. (2020) and Hwang et al. (2021), the unimodal posteriors are fused to obtain a joint posterior. Subsequent generations rely on sampling from this joint posterior, which becomes the primary optimization target. As a result, Hwang et al. (2021) suggests that the asymmetry of the KL divergence makes the forward KL more suitable than the reverse KL for this context.

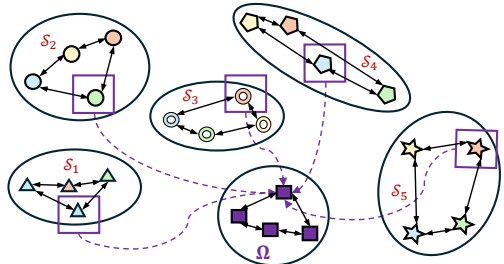

Figure 8: **Visualization of the effects of the two regularization terms on a five-view sample with one missing view.** Each circle represents a set of homogeneous latent variables. In $\mathcal{S}_l$, markers of the same shape indicate variables from the $l$-th view, while colors represent their source views. In $\Omega$, consensus variables are fused from all five views, as shown by the dashed purple arrows. Bidirectional black arrows illustrate the cyclic convergence of variables within each set.

In contrast, our approach maintains the unimodal posteriors throughout the factorization process, with subsequent reconstructions dependent on these individual subspaces. Thus, optimizing all unimodal posteriors becomes essential. The symmetric KL divergence ensures that both distributions are encouraged to move toward each other's high-probability regions, fostering more stable convergence during training.

### A.5 LATENT SPACE DYNAMICS DURING TRAINING

The two regularization terms, **Inter-View Translatability** and **Consensus Concentration**, play distinct roles in the training process. These terms impact the arrangement and interaction of latent variables in the learned space, as illustrated in the following visualizations.

In Figure 8, we depict the impact of the regularization terms on a five-view sample with one missing view. In the single-view cell $\mathcal{S}_l$, markers of the same shape represent the latent variables $z$ corresponding to the $l$-th view, while the different colors indicate their source views, whether self-encoded or cross-transformed. Note that each set contains as many variables as there are observed views (four in this case), as they can only be encoded or transformed from available views. As this term diminishes, the variables within each $\mathcal{S}_l$ cyclically converge, indicating that variables from different views can effectively transform into each other, thereby establishing inter-view correspondences. This process also enforce a form of soft consistency, as representations from different views are encouraged to approach each other after being transformed, rather than aligning directly. The learning of inter-view correspondences avoids collapsing into identity mappings because the reconstruction loss ensures that variables retain unique information specific to each view.

The Consensus Concentration term aims to ensure that consensus variables derived from different combinations remain consistent, as shown in Figure 8. Each $\omega$ in $\Omega$ is obtained from a complete combination of all views. Over time, the regularization promotes closer alignment of these consensus variables, facilitating the aggregation of shared information across the views.

Figure 9 illustrates the evolution of the latent space during training. The top row depicts the latent variables $z$ for five views across different training epochs, with each color representing a different view. Each colored cluster contains all variables in $\mathcal{S}_l$, whether self-encoded or cross-transformed. Initially, these variables are scattered, but over time, they coalesce into five distinct clusters, indicating the emergence of inter-view correspondences. In the middle row, the consensus variables $\omega$, derived from various complete combinations of the five views, are shown. At the start, these variables are widely dispersed, as the views are not yet able to transform into each other effectively, leading to inconsistencies in the captured information. As training progresses, inter-view correspondences are established, and the first $k$ dimensions of $z$ reliably encode shared information across views. As a result, all combinations of the five views produce similar $\omega$'s, with their representations converging into indistinguishable, uniformly distributed clusters.

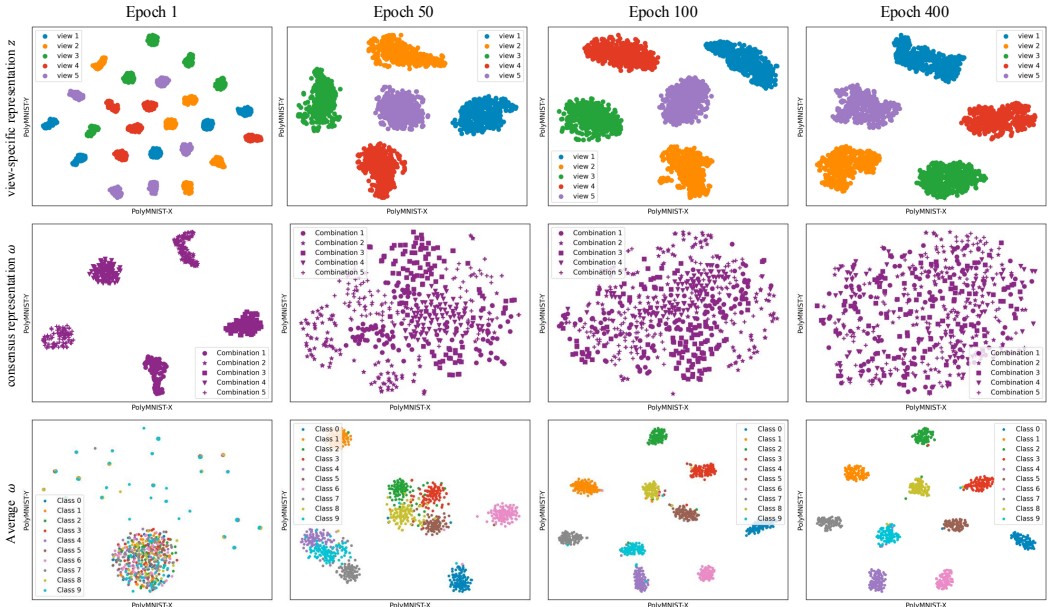

Figure 9: **T-SNE visualization of latent space dynamics during training on the PolyMNIST dataset. Top row**: Latent variables $z$ for five views at different training stages, with colors representing each view. As training progresses, the initially scattered representations gradually cluster, indicating the establishment of inter-view correspondences. **Middle row**: Consensus variables $\omega$, derived from different combinations of the five views, are shown with different marker shapes. These scattered representations gradually align and become more consistent. **Bottom row**: Average consensus representations, colored by digit class, become more distinct over time, which reflects enhanced clustering and effective information sharing across views.

## B    ADDITIONAL EXPERIMENTAL RESULTS

In this section, we present additional experimental results to complement those in the main text. A complete version of the clustering results on five datasets in Section 4.1, including standard deviations from five experimental runs, is provided in Table 3.

### B.1    QUANTITATIVE AND QUALITATIVE RESULTS ON MVSHAPENET

In this section, we present both quantitative and qualitative comparisons on the MVShapeNet dataset to assess the performance of our method, MVP, alongside several prominent MVAE-based approaches. These include the models discussed in Section 4.2.2, such as mVAE, mmVAE, mmJSD, MoPoE, and MVTCAE. Although MMVAE+ is a strong method, it did not perform well on this dataset using the same CNN-based architecture, irrespective of whether the Normal or Laplace distribution was applied. For this reason, we chose not to include it in our direct comparisons.

It's important to note that MMVAE+ excels in handling complete datasets by leveraging auxiliary priors to facilitate cross-modal reconstructions, making it highly effective in real-world generation tasks. However, its design, which prioritizes robustness to hyperparameters that control the capacity of modality-specific subspaces, seems less suited for scenarios where data is missing. In such cases, its auxiliary prior may struggle to compensate for missing information, affecting its ability to perform well under these conditions. This key distinction highlights the difference in focus between MMVAE+ and our approach, with each serving different objectives.

Figure 10 illustrates multi-view sample generation conditioned on view 5 across different object categories. All models were trained with a missing rate of $\eta = 0.5$, representing a complex scenario where maintaining geometric consistency and capturing fine details across views is particularly challenging. MVP stands out by producing sharper and more consistent multi-view samples across categories compared to the other models.

Table 3: Complete clustering results of nine methods on five multi-view datasets with missing rates of $\eta = 0.1, 0.3, 0.5$, and 0.7. The first and second best results are indicated in **bold red** and **blue**, respectively. Each experiment was run five times using different random seeds.

| Dataset | Metrics | 0.1 ACC↑ | 0.1 NMI↑ | 0.1 ARI↑ | 0.3 ACC↑ | 0.3 NMI↑ | 0.3 ARI↑ | 0.5 ACC↑ | 0.5 NMI↑ | 0.5 ARI↑ | 0.7 ACC↑ | 0.7 NMI↑ | 0.7 ARI↑ |
|---|---|---|---|---|---|---|---|---|---|---|---|---|---|
| Handwritten | DCCA | 73.83±0.66 | 71.17±1.92 | 55.13±2.01 | 68.02±5.87 | 64.93±6.40 | 44.76±11.45 | 63.25±8.30 | 60.27±8.42 | 38.93±12.49 | 59.07±10.23 | 56.44±9.86 | 33.58±14.25 |
| | DCCAE | 73.61±0.65 | 72.22±1.62 | 54.61±2.49 | 68.12±5.53 | 65.76±6.62 | 43.42±12.07 | 63.36±8.13 | 60.72±8.97 | 37.32±13.48 | 59.31±9.94 | 56.89±10.22 | 32.55±14.33 |
| | DIMVC | **89.13±1.11** | 80.06±1.36 | 77.96±2.06 | 85.24±1.20 | 74.67±1.47 | 70.83±2.10 | 82.76±0.71 | 71.17±1.06 | 66.81±1.19 | 79.66±4.28 | 68.94±2.20 | 63.12±4.17 |
| | DSIMVC | 81.27±1.48 | 79.47±1.30 | 71.59±1.92 | 81.82±3.27 | 80.27±2.95 | 73.36±3.40 | 81.39±2.83 | 79.23±2.47 | 71.88±3.94 | 77.38±4.35 | 74.80±3.12 | 66.84±4.35 |
| | Completer | 82.18±2.06 | 77.73±1.03 | 68.92±1.98 | 78.70±4.22 | 69.08±4.17 | 58.14±4.50 | 74.73±4.51 | 67.49±4.05 | 50.21±6.37 | 68.86±2.41 | 62.41±1.61 | 41.06±1.88 |
| | CPSPAN | 80.30±7.13 | 78.43±3.90 | 71.84±6.71 | 79.80±6.76 | 79.08±3.61 | 72.28±6.09 | 84.64±5.47 | 80.23±3.10 | 75.34±5.62 | **83.90±5.26** | 80.60±1.98 | **75.25±4.26** |
| | ICMVC | 88.86±5.39 | 82.19±3.94 | 79.51±6.58 | 80.95±2.75 | 74.53±1.15 | 68.15±1.81 | 73.83±0.46 | 67.95±0.23 | 58.72±0.44 | 67.48±3.83 | 61.99±2.97 | 48.66±3.19 |
| | DVIMC | 87.89±3.28 | **83.51±1.98** | **79.66±2.81** | 85.36±5.55 | **82.82±3.82** | **78.50±5.83** | **85.01±4.49** | **82.96±2.41** | **78.50±4.06** | 81.66±5.92 | **80.78±1.84** | 74.85±4.66 |
| | MVP(Ours) | **90.55±5.29** | **87.08±2.48** | **84.46±5.16** | **88.69±5.89** | **84.86±2.73** | **81.59±5.77** | **90.76±4.69** | **85.44±1.55** | **83.53±3.69** | **86.74±5.15** | **82.56±2.31** | **78.75±4.77** |
| CUB | DCCA | 58.93±3.21 | 59.59±1.08 | 40.87±1.23 | 55.65±4.49 | 54.23±5.71 | 35.63±5.86 | 48.60±10.85 | 46.71±11.92 | 27.88±12.17 | 42.35±14.35 | 39.42±16.37 | 21.77±14.94 |
| | DCCAE | 57.27±3.22 | 61.04±2.59 | 43.09±3.32 | 53.43±4.53 | 54.59±6.82 | 36.48±7.11 | 47.21±9.68 | 47.05±12.10 | 28.37±12.93 | 41.52±13.12 | 40.50±15.76 | 22.66±15.03 |
| | DIMVC | 66.03±5.04 | 61.70±4.06 | 48.96±5.55 | 57.20±3.98 | 56.00±2.89 | 41.25±3.74 | 60.65±3.62 | 55.75±3.62 | 42.81±4.77 | 56.08±1.16 | 51.07±1.54 | 36.65±2.23 |
| | DSIMVC | **72.93±5.28** | **67.82±3.07** | **55.89±4.56** | **66.83±4.74** | 61.78±2.21 | 47.71±3.84 | **68.37±5.67** | 61.55±3.00 | **48.21±4.70** | **67.33±2.29** | 59.89±0.95 | **46.31±1.97** |
| | Completer | 52.97±4.56 | 65.47±2.41 | 45.98±4.45 | 60.73±2.49 | **68.88±1.94** | **52.78±2.71** | 51.90±2.47 | 61.84±0.89 | 45.11±1.28 | 19.43±1.15 | 17.37±1.23 | 0.73±0.31 |
| | CPSPAN | 58.77±3.12 | 62.27±2.18 | 45.35±2.40 | 61.30±1.41 | 64.21±2.32 | 48.93±2.94 | 60.07±4.51 | **64.18±2.50** | 46.42±3.41 | 58.60±1.41 | **62.16±1.43** | 45.23±1.64 |
| | ICMVC | 29.23±0.37 | 38.31±1.87 | 21.55±0.88 | 24.33±4.17 | 25.74±5.40 | 14.17±4.28 | 22.90±2.83 | 19.87±1.93 | 10.52±1.76 | 19.43±2.62 | 16.10±1.80 | 7.79±1.67 |
| | DVIMC | 44.53±4.00 | 41.83±3.49 | 23.78±2.59 | 43.37±2.57 | 45.18±1.85 | 28.29±1.80 | 39.57±5.54 | 34.39±4.68 | 20.52±3.40 | 39.47±2.61 | 36.71±3.60 | 22.41±2.82 |
| | MVP(Ours) | **78.67±2.39** | **77.67±0.66** | **66.73±0.71** | **74.97±4.58** | **73.09±2.55** | **61.35±3.60** | **74.20±3.21** | **71.40±1.30** | **59.11±2.53** | **66.53±3.73** | **63.11±1.21** | **50.59±1.84** |
| Scene15 | DCCA | 38.22±1.00 | 41.20±0.52 | 19.89±0.42 | 36.16±2.17 | 39.46±1.79 | 17.12±2.80 | 34.05±3.50 | 37.26±3.49 | 14.48±4.45 | 30.84±6.35 | 33.93±6.55 | 12.60±5.08 |
| | DCCAE | 39.46±0.84 | 42.08±0.55 | 20.36±0.27 | 36.73±2.86 | 39.80±2.35 | 17.06±3.32 | 34.49±3.96 | 37.66±3.61 | 14.57±4.44 | 31.16±6.75 | 34.21±6.84 | 12.64±5.13 |
| | DIMVC | 42.51±2.42 | 41.53±1.60 | 24.45±2.23 | 40.37±1.85 | 38.57±1.30 | 20.84±1.54 | 40.17±2.14 | 35.95±2.50 | 20.59±2.68 | 36.01±1.27 | 32.57±1.08 | 16.29±2.24 |
| | DSIMVC | 29.43±1.21 | 30.38±0.95 | 14.86±0.59 | 31.38±1.02 | 32.54±1.33 | 16.29±0.90 | 27.24±1.21 | 28.68±0.85 | 13.38±0.64 | 28.42±0.78 | 29.09±0.81 | 13.85±0.21 |
| | Completer | 37.00±1.82 | 41.89±0.41 | 23.60±0.84 | 40.04±0.67 | 42.41±0.32 | 24.22±0.24 | 36.64±1.91 | 38.99±1.05 | 19.70±1.35 | 35.37±0.87 | 37.05±1.03 | 17.58±0.99 |
| | CPSPAN | 42.69±1.77 | 38.79±2.49 | 24.56±2.15 | 43.21±1.51 | 39.42±0.56 | 24.94±0.85 | 43.44±1.43 | 39.19±1.85 | 24.96±1.46 | **42.53±2.56** | **38.41±2.43** | **24.38±1.97** |
| | ICMVC | 43.88±2.35 | 40.03±1.14 | 25.80±1.54 | 43.14±1.02 | 38.06±0.51 | 24.74±0.89 | 37.96±1.87 | 33.45±0.93 | 20.34±0.78 | 36.70±2.22 | 35.80±1.30 | 18.35±1.32 |
| | DVIMC | **45.16±2.40** | **45.06±1.33** | **28.64±1.52** | **43.68±1.54** | **42.32±2.11** | **26.68±1.05** | **41.13±3.32** | **39.58±2.25** | **25.03±2.42** | 39.59±4.41 | 36.66±4.62 | 21.56±4.04 |
| | MVP(Ours) | **45.70±1.63** | 43.77±0.93 | 27.90±1.38 | **45.81±2.75** | **42.54±1.02** | **27.53±1.86** | **45.28±1.44** | **41.84±0.79** | **26.80±1.20** | **43.14±2.20** | **39.53±0.67** | **24.68±1.58** |
| Reuters | DCCA | 47.66±1.83 | 23.93±4.52 | 15.46±1.55 | 46.28±1.95 | 20.62±4.64 | 12.71±3.05 | 44.10±4.44 | 22.63±4.89 | 11.04±3.63 | 43.36±4.35 | 22.90±5.13 | 10.03±3.69 |
| | DCCAE | 42.70±1.20 | 23.84±6.27 | 7.59±1.61 | 43.71±2.93 | 26.07±5.37 | 8.15±2.47 | 42.32±3.13 | 24.30±6.08 | 6.80±2.78 | 41.32±3.25 | 23.11±6.34 | 5.90±2.88 |
| | DIMVC | 48.83±2.38 | 28.94±2.39 | 25.78±2.01 | 50.54±2.91 | 29.86±2.58 | **26.89±1.90** | 48.51±2.54 | 27.29±2.25 | 24.74±1.67 | 46.94±3.41 | 25.79±2.77 | 23.24±2.49 |
| | DSIMVC | 51.26±3.45 | 35.56±1.95 | 28.21±2.05 | **51.33±2.28** | **34.88±1.00** | 26.61±1.81 | **50.78±2.16** | **36.85±1.32** | **28.27±0.81** | **47.12±2.08** | **33.57±3.00** | **25.51±1.96** |
| | Completer | 41.08±0.97 | 21.38±4.42 | 7.92±2.18 | 40.56±2.08 | 22.48±1.97 | 10.32±2.34 | 41.77±2.28 | 20.41±2.78 | 9.80±3.53 | 42.27±2.73 | 22.47±1.17 | 11.51±1.84 |
| | CPSPAN | 38.35±5.07 | 14.35±3.10 | 10.94±2.70 | 38.51±2.30 | 13.11±4.75 | 10.47±2.09 | 38.21±3.44 | 11.80±3.58 | 11.30±3.35 | 37.86±4.66 | 12.03±4.50 | 10.16±3.84 |
| | ICMVC | **54.01±1.67** | **36.52±1.37** | **29.44±1.13** | 51.09±2.33 | 30.71±2.09 | 25.66±2.38 | 47.59±1.68 | 28.43±1.00 | 23.56±2.07 | 47.67±0.47 | 26.83±1.54 | 22.14±1.44 |
| | DVIMC | 44.06±1.87 | 16.08±5.59 | 15.21±4.70 | 43.06±1.02 | 10.84±0.61 | 11.77±1.06 | 35.37±3.99 | 5.14±2.33 | 4.98±2.51 | 32.18±4.07 | 3.02±2.48 | 3.15±2.72 |
| | MVP(Ours) | **57.83±3.66** | **37.25±1.17** | **32.20±2.02** | **55.70±5.10** | **37.02±2.08** | **31.35±3.30** | **53.67±2.88** | **35.43±1.16** | **30.24±0.93** | **55.16±2.93** | **36.00±0.51** | **30.66±1.07** |
| SensIT Vehicle | DCCA | 57.11±5.77 | 11.60±8.78 | 14.26±11.24 | 57.76±5.86 | 14.46±11.04 | 16.62±13.17 | 53.89±8.00 | 11.01±10.83 | 12.79±12.80 | 50.69±9.05 | 8.47±10.37 | 9.75±12.28 |
| | DCCAE | 57.93±5.13 | 12.84±7.71 | 15.28±10.61 | 60.32±4.77 | 19.42±9.94 | 22.46±12.31 | 54.08±9.96 | 13.32±11.89 | 15.40±14.21 | 51.33±9.86 | 10.31±11.55 | 11.81±13.79 |
| | DIMVC | 59.72±8.27 | 17.31±13.96 | 21.82±8.45 | 62.38±5.96 | 23.18±10.38 | 27.93±12.98 | 61.09±6.02 | 22.08±11.01 | 26.21±13.13 | 60.57±4.44 | 21.36±9.41 | 25.44±11.21 |
| | DSIMVC | 69.82±1.60 | 33.40±0.62 | 34.88±3.00 | 69.24±0.98 | **32.95±0.41** | 33.50±1.66 | **68.05±0.85** | **31.49±0.23** | 31.56±1.61 | **66.54±0.22** | **30.08±0.22** | 29.73±0.72 |
| | Completer | 52.63±2.56 | 5.33±1.96 | 3.72±3.01 | 55.59±5.66 | 12.09±11.52 | 11.29±12.69 | 55.09±6.18 | 13.96±11.70 | 12.52±12.63 | 56.37±6.36 | 14.77±11.10 | 14.66±12.71 |
| | CPSPAN | 63.48±1.65 | 28.43±0.41 | 32.32±0.59 | 64.03±1.24 | 28.10±0.32 | 22.33±0.84 | 65.47±0.89 | 28.62±60.78 | 32.25±0.55 | 64.16±1.76 | 28.60±0.79 | **31.28±0.79** |
| | ICMVC | **71.50±0.46** | **34.53±0.38** | **36.41±0.42** | **70.79±0.50** | 32.95±0.59 | **35.26±0.38** | 67.80±2.36 | 29.11±2.47 | 29.36±2.48 | 54.11±5.51 | 19.39±3.44 | 18.93±3.51 |
| | DVIMC | 69.48±0.46 | 30.41±0.52 | 34.98±0.88 | 69.58±0.21 | 30.31±0.23 | 33.63±0.43 | 67.89±0.46 | 29.27±0.26 | **34.00±0.43** | 61.91±2.77 | 25.84±0.90 | 28.59±2.35 |
| | MVP(Ours) | **72.08±0.10** | **34.81±0.21** | **41.10±0.15** | **71.28±0.23** | **33.57±0.09** | **39.76±0.26** | **70.21±0.09** | **32.05±0.20** | **38.06±0.12** | **68.87±0.32** | **30.08±0.09** | **36.23±0.39** |

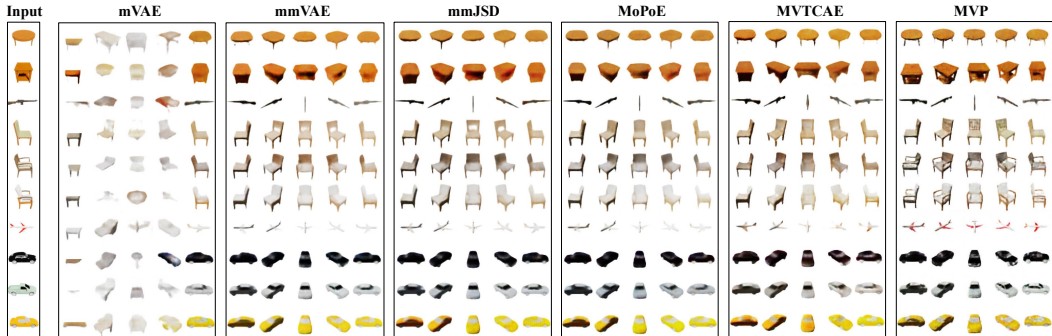

Figure 10: **Multi-view sample generation conditioned on view 5.** The leftmost column shows input images from view 5, randomly selected from five categories: table, rifle, chair, airplane, and car. The following columns display five-view samples generated by different models.

Table 4: **Quantitative results on the MVShapeNet dataset.** The metrics 'ACC' and 'View' represent the classification accuracy of generated images for object categories and perspective angles, respectively, using two pretrained classifiers. The metric 'SSIM' measures the structural similarity of generated images compared to the ground truth. Models were trained with incomplete data at missing rates $\eta = 0.1$ and $\eta = 0.5$, and evaluated across different combinations of missing views. Results are averaged over same-sized subsets such as 'missing 1 view', which includes five cases: {2,3,4,5}, {1,3,4,5}, {1,2,4,5}, and {1,2,3,4}.

| Testing Subset | | Missing 1 view | | | Missing 2 views | | | Missing 3 views | | | Missing 4 views | | |
|---|---|---|---|---|---|---|---|---|---|---|---|---|---|
| | Method | ACC↑ | View↑ | SSIM↑ | ACC↑ | View↑ | SSIM↑ | ACC↑ | View↑ | SSIM↑ | ACC↑ | View↑ | SSIM↑ |
| 0.1 | mVAE | 90.56 | 81.34 | 0.8406 | 86.60 | 78.56 | 0.8139 | 79.11 | 73.97 | 0.7737 | 62.27 | 66.70 | 0.7045 |
| | mmVAE | 91.56 | 81.28 | 0.8141 | 91.55 | 81.39 | 0.8132 | 91.56 | 81.42 | 0.8126 | 91.78 | 81.51 | 0.8141 |
| | mmJSD | 41.16 | 53.03 | 0.7217 | 40.91 | 53.11 | 0.7210 | 40.77 | 53.24 | 0.7207 | 40.55 | 53.16 | 0.7215 |
| | MoPoE | 86.45 | 83.18 | 0.8252 | 86.34 | 83.04 | 0.8223 | 85.94 | 82.66 | 0.8159 | 84.40 | 81.68 | 0.7961 |
| | MVTCAE | 93.65 | 83.45 | **0.8570** | 93.39 | 83.41 | **0.8440** | 92.71 | 83.07 | **0.8308** | 89.90 | 81.70 | 0.7858 |
| | **MVP(Ours)** | **94.66** | **84.34** | 0.8349 | **94.76** | **84.35** | 0.8336 | **94.77** | **84.58** | 0.8274 | **94.47** | **84.12** | **0.8259** |
| 0.5 | mVAE | 79.04 | 76.14 | 0.8111 | 73.02 | 71.88 | 0.7802 | 63.76 | 66.88 | 0.7386 | 49.28 | 61.26 | 0.6800 |
| | mmVAE | 91.22 | 81.67 | 0.8088 | 91.26 | 81.63 | 0.8081 | 91.30 | 81.51 | 0.8076 | 91.38 | 81.40 | 0.8092 |
| | mmJSD | 92.40 | 82.49 | 0.8118 | 92.41 | 82.46 | 0.8110 | 92.40 | 82.52 | 0.8105 | 92.37 | 82.47 | **0.8192** |
| | MoPoE | 92.84 | 82.46 | 0.8314 | 92.71 | 82.31 | 0.8292 | 92.26 | 82.00 | 0.8247 | 90.55 | 80.91 | 0.8129 |
| | MVTCAE | 93.38 | 83.09 | **0.8482** | 93.07 | 82.91 | **0.8396** | 92.22 | 82.53 | **0.8540** | 89.03 | 80.86 | 0.7865 |
| | **MVP(Ours)** | **94.57** | **84.60** | 0.8239 | **94.72** | **84.64** | 0.8276 | **94.75** | **84.66** | 0.8268 | **94.45** | **84.50** | 0.8126 |

Table 4 presents the quantitative evaluation on the MVShapeNet dataset. The 'ACC' and 'View' metrics represent classification accuracy for object categories and perspective angles, respectively, while 'SSIM' measures the structural similarity between generated and ground truth images. As seen from the visualization, SSIM primarily captures the basic structure of the generated images, resulting in relatively minor differences across most methods. Our method, MVP, consistently achieves competitive SSIM scores and surpasses all other models in both classification accuracy metrics. This robustness is particularly evident compared to mVAE, which suffers a notable drop in performance as the number of missing views increases. Figure 10 illustrates this, showing how mVAE, relying on PoE fusion, loses the structural integrity of the input object when fewer views are available.

Interestingly, mVAE, mmJSD, and MoPoE perform better in scenarios with higher missing rates. This can be attributed to their reliance on fusing all available views in their posterior or informational priors, making them dependent on missing data to memorize different incomplete combinations. Thus, they struggle to fully utilize the complementary information provided by additional views. On the other hand, mmVAE, which uses MoE fusion, shows improved performance with more available views during both training and testing. However, its fusion strategy struggles to effectively aggregate information from multiple views, resulting in smoother generated objects that may lack the distinct angles necessary to capture fine details.

MVTCAE demonstrates stable performance across all conditions by enforcing strict consistency between views, making it resilient to missing views. However, this strict consistency can result in the loss of unique, view-specific details. Our MVP method, by enforcing consistency after trans-

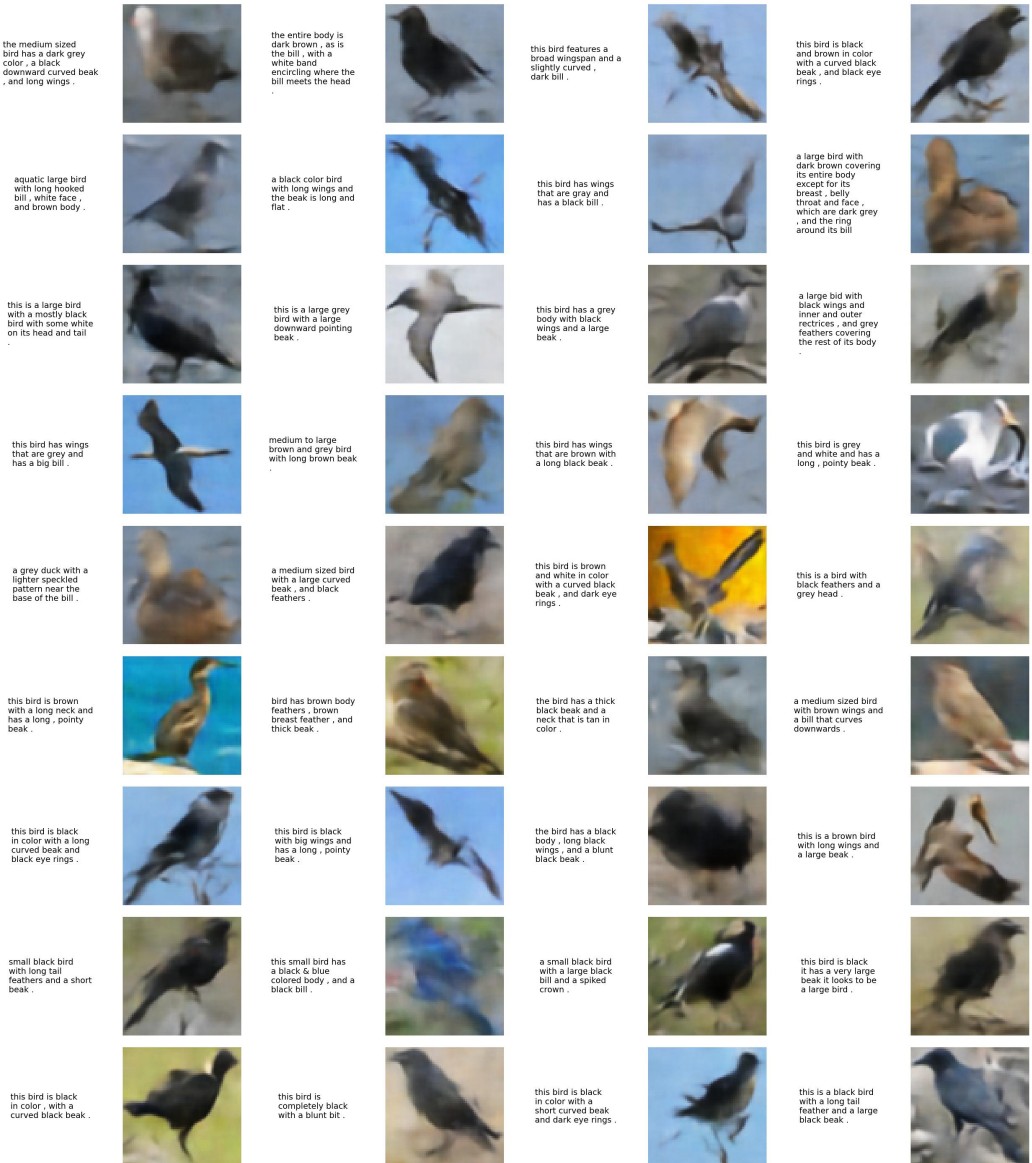

Figure 11: Conditional generation by our method on the CUB dataset given only text.

formations, not only maintains robust performance but also effectively integrates information from additional views while preserving the sharp, distinctive details of each view.

## B.2 ADDITIONAL EXPERIMENT RESULTS ON CUB DATASET

We conducted additional experiments on the raw-text version of the CUB dataset to evaluate the generative capabilities of our method on datasets with non-RGB views (Netzer et al., 2011; Shi et al., 2019). This version contains 88,550 training and 29,330 testing samples, comprising paired bird images and textual descriptions. Following MMVAE+ (Palumbo et al., 2023), we adopted the same network architecture and latent dimension (64 dimensions).

The results, shown in Figure 10, demonstrate that our method effectively generates images aligned with textual descriptions in the incomplete setting. Clear semantic alignments are observed in attributes such as colors (e.g., black, white, brown) and structures (e.g., belly, beak, wings). However, the generated images exhibit blurry backgrounds and contours, consistent with observations in MM-

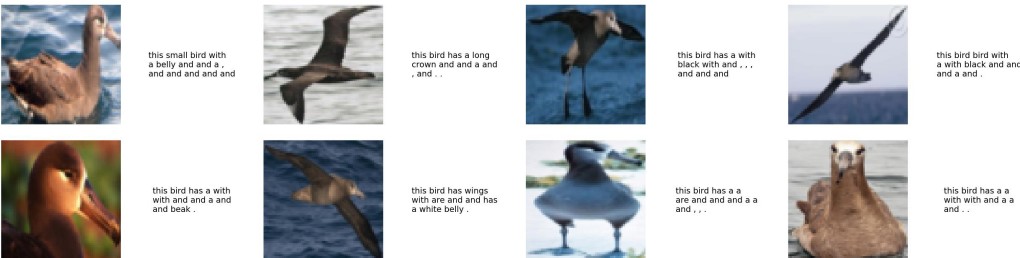

Figure 12: Conditional generation by our method on the CUB dataset given only images.

Table 5: **Ablation study of permutation and different priors in MVAEs with inter-view correspondences.** Clustering results on the Handwritten dataset with a missing rate of 0.5, averaged over five training runs. "Perm." refers to the use of permutation, which reorders the variables of the same views (indicated by the same superscript) for reconstruction. The "Regularization" column indicates the use of different prior settings. The last row represents our proposed model.

| Model | Reconstruction | Regularization | ACC↑ | NMI↑ | ARI↑ |
|---|---|---|---|---|---|
| 1 | Random Perm. | Diagonal | 31.58±10.75 | 30.98±12.11 | 16.94±8.81 |
| 2 | Random Perm. | Random | 59.37±4.30 | 58.70±2.73 | 44.69±4.33 |
| 3 | w/o Perm. | Cyclic | 79.80±1.26 | 81.35±1.00 | 73.98±1.46 |
| 4 | Cyclic Perm. | $\mathcal{N}(0,1)$ | 86.96±6.44 | 82.49±2.80 | 79.23±5.65 |
| 5 | Cyclic Perm. | Fusion | 87.71±6.09 | 84.47±1.57 | 80.77±4.61 |
| 6 | Cyclic Perm. | Diagonal | 88.53±6.43 | 85.00±2.86 | 81.77±6.14 |
| Ours | Cyclic Perm. | Cyclic | **90.76±4.69** | **85.44±1.55** | **83.53±3.69** |

VAE+, which stem from the limitations of single-step VAE-based generation. Advanced approaches, such as D-CMVAE (Palumbo et al., 2024), address this issue using diffusion models, though such refinements are beyond the scope of this study.

Quantitative evaluation remains challenging, as traditional metrics like diversity or reconstruction scores fail to capture cross-modal consistency. While MMVAE+ introduced a metric based on HSV color alignment with textual descriptions, it does not measure higher-level consistencies, such as specific bird features (e.g., wings, beak) or environmental contexts (e.g., sky, water). Future directions could explore more precise metrics or employ large vision-language models for automated evaluation (Lin et al., 2025).

### B.3 ABLATION STUDY OF PERMUTATION AND PRIORS SETTING

We conducted an ablation study to validate the key design decisions in our method, and as shown in Table 5, our proposed approach achieves the best performance. For MVAEs that explicitly model inter-view correspondences, given incomplete input data $\{x^{(v)}\}_{v\in\mathcal{I}}$, we derive a set of latent variables $\{z_v^{(l)}\}$ corresponding to different views, organized into matrix $Z_0$ for clarity. In this notation, the superscript in $z_v^{(l)}$ indicates that it represents the $l$-th view, while the subscript denotes the source view. Thus, $z_v^{(l)}$ is used to reconstruct the $l$-th view's observation and can be regularized by other latent variables associated with the $l$-th view, all of which share the same superscript in the $l$-th single-view cell $\mathcal{S}_l = \{z_v^{(l)}\}_{v\in\mathcal{I}}$.

**Model 1** serves as a simple baseline. In this model, we randomly select a latent variable from the $l$-th view to reconstruct the observation $x^{(v)}$ for that view, using $z_l^{(l)}$ (with matching superscripts and subscripts) to regularize all variables for the $l$-th view. However, this approach performs poorly because the random selection can pick either the self-view encoded $z$ ($v = l$) or the cross-view transformed $z$ ($v \neq l$), leading to difficulties in reconstruction and confusing the learning process. Additionally, the regularization term for $z_l^{(l)}$ simply vanishes entirely.

**Model 2** relaxes the prior by replacing the strict $z_l^{(l)}$ with a posterior derived from a random permutation. This improves performance due to the added randomness. However, since random permutations do not guarantee that elements are moved from their original positions, some regularization terms for $z_v^{(l)}$ may still vanish.

In **Model 3**, we exclude permutation during reconstruction. For the $l$-th view $x^{(l)}$, we encode it into $z_l^{(l)}$ and use it directly for decoding. Other latent variables, derived through inter-view correspondences, are aligned with the target view's variable solely through regularization terms. This approach slightly improves performance, as it effectively adds information from other views during decoding. However, it cannot establish and learn correspondences between views effectively.

In **Models 4-6** and our proposed model, we apply cyclic permutation to the matrix $Z_0$, using both the diagonals of matrices $Z_0$ and $Z_1$ for reconstruction, effectively combining self-view reconstruction with cross-view generation. The use of different informational priors ('Fusion', 'Diagonal', and 'Cyclic') demonstrates clear advantages over the standard normal distribution $\mathcal{N}(0, 1)$, which lacks the flexibility to adapt to varying samples.

- **'Fusion'** involves using the geometric mean of posteriors within a homogeneous set as the prior for each posterior in that set, it is kind of like reducing the variance of all unimodal posteriors. For example, the geometric mean of $\{z_v^{(l)}\}_{v \in \mathcal{I}}$ is used to regularize each $z_v^{(l)}$. However, when views are missing, the varying set sizes $|\mathcal{I}|$ for different samples result in imbalanced fusion and increased computational complexity.

- **'Diagonal'** regularizes all $\{z_v^{(l)}\}_{v \in \mathcal{I}}$ using the distribution of $z_l^{(l)}$, and the geometric mean of $\{z_l^{(l)}\}_{l \in \mathcal{I}}$ to regularize all $\omega$. However, when $v = l$, the regularization term fails, leading to suboptimal performance.

- **'Cyclic'**, as used in our method, efficiently reuses the permuted posteriors as priors and fully leverages the relationships between views. Since no element remains in its original position, this method maximizes inter-view correspondences and achieves the best performance. Additionally, it avoids extra fusion computation for informational priors by simply calculating the KL divergence of $Z_0$ and $Z_1$.

In all, our model outperforms all others, demonstrating the effectiveness of our learning strategy.

## B.4 COEFFICIENTS $\beta$ OF REGULARIZATION TERMS

As is common in the VAE literature, the objective function in Eq. (4) can be rewritten as the sum of a reconstruction term and two KL-divergence terms, each weighted by the coefficients $\beta_1$ and $\beta_2$ (Higgins et al., 2017). As shown in Figure 13, we conduct a sensitivity analysis on the coefficients to examine how different combinations of $\beta_1$ (for $z$ regularization) and $\beta_2$ (for $\omega$ regularization) impact clustering accuracy. Since $\omega$ is deterministically computed from fusion of $z$, both terms essentially act as regularization on $z$, but with different objectives. The figure shows that the best performance is achieved when $\beta_1 = 5.0$ and $\beta_2 = 2.5$, resulting in a clustering accuracy of 90.76. This indicates that a moderate weighting of both regularization terms strikes an optimal balance between promoting smooth latent space transitions (**Inter-View Translatability**) and ensuring latent space consistency (**Consensus Concentration**).

Interestingly, the performance degrades when either $\beta_1$ or $\beta_2$ is set too high, as seen when $\beta_1 = 10$ or $\beta_2 = 10$. This likely results from overly constraining the latent space, which could reduce the flexibility needed for effective inter-view transformations. Conversely, lower values of $\beta_1$ and $\beta_2$ (e.g., $\beta_1 = 1.0$, $\beta_2 = 1.0$) show suboptimal performance, indicating insufficient regularization and thus poorer structure in the latent space.

## B.5 THE CHOICE OF LATENT REPRESENTATION DIMENSIONS

In this section, we examine the choice of latent variable dimensions across different datasets and compare the selection of the first $k$ dimensions for encoding shared information in the PolyMNIST and MVShapeNet datasets. For the clustering task, since it requires high consistency between views to preserve shared information and reveal the underlying category structure, we use 100% of the

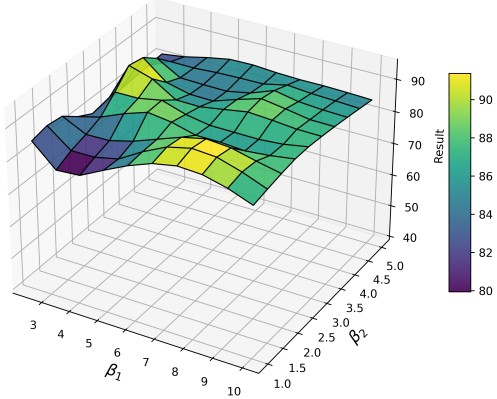

Figure 13: **Sensitivity analysis of the coefficients for the two regularization terms in the ELBO.**
$\beta_1$ controls the regularization for $z$ and $\beta_2$ controls the regularization for $\omega$. Clustering accuracy (ACC) is reported on the Handwritten dataset with missing rate 0.5, averaged over five training runs.

Table 6: **Clustering results across different representation dimensions** ($z$). Clustering accuracy (mean ± standard deviation) is shown for various latent dimensions with a missing rate of $\eta = 0.5$, averaged over five runs. The green cell marks the selected dimension.

| Dataset | Dimension of latent representation | | | |
|---|---|---|---|---|
| **Handwritten** | **8** | **16** | **24** | **32** |
| | 85.88±6.15 | 90.76±4.69 | 88.32±5.99 | 86.41±5.63 |
| **CUB** | **8** | **16** | **32** | **64** |
| | 67.30±5.36 | 74.20±3.21 | 73.03±3.97 | 69.40±2.95 |
| **Scene15** | **8** | **16** | **32** | **64** |
| | 42.58±1.31 | 45.28±1.44 | 44.44±0.69 | 43.74±1.79 |
| **Reuters10** | **8** | **16** | **32** | **48** |
| | 52.60±2.66 | 53.67±2.88 | 53.89±5.32 | 53.01±4.26 |
| **SensIT Vehicle** | **8** | **16** | **32** | **64** |
| | 70.33±0.48 | 69.95±0.28 | 70.21±0.09 | 70.00±0.21 |

dimensions of $z$ to encode shared information, meaning $k = d$, with $\omega$ having the same dimensionality as $z$. Table 6 presents the clustering results for various latent representation dimensions across different datasets. Rather than always selecting the best-performing dimension, we strike a balance between model complexity and the inherent structure of the dataset for simplicity.

Table 7 reports the reconstruction performance on the PolyMNIST and MVShapeNet datasets as we vary the proportion of dimensions ($k$) used to encode shared information. The results show that performance remains robust across different ratios, allowing us to select the ratio based on each dataset's characteristics. For PolyMNIST, which exhibits greater variability between views due to diverse background styles and colors, a smaller $k/d$ ratio (50%) for shared information encoding is more effective. This is because only a small portion of the pixel data (representing the digit) is consistent across views, while the rest varies significantly. In contrast, MVShapeNet, with more consistent views (mainly different angles of the same object against a plain background), is better suited to a higher $k/d$ ratio (75%).

Table 7: **Reconstruction results across different shared feature dimensions** ($k$) **in the** $d$**-dimensional** $z$**.** The models are trained on the PolyMNIST and MVShapeNet datasets with a missing rate of $\eta = 0.5$. SSIM is reported on the complete test set. The green cell highlights the selected dimension.

| k/d | d | 25% | 50% | 75% | 100% |
|---|---|---|---|---|---|
| PolyMNIST | 96 | 0.6023 | 0.6188 | 0.6137 | 0.6134 |
| MVShapeNet | 256 | 0.8383 | 0.8384 | 0.8381 | 0.8374 |

## C  Implementation Details of Our Method

### C.1  Network Architectures

For the experiments in Section 4.1, we employed fully connected neural networks similar to those used in previous studies. The choice of architecture depends on the dataset's characteristics, such as input dimension and number of samples. We used one of the following network configurations:

- $d_v$ - 256 - 256 - 1024 - $d$
- $d_v$ - 512 - 512 - 1024 - $d$
- $d_v$ - 512 - 512 - 2048 - $d$
- $d_v$ - 1024 - 1024 - 2048 - $d$

Here, $d_v$ is the input dimension, and $d$ is the latent dimension. Each fully connected layer is followed by a ReLU activation function to introduce non-linearity, except for the final output layer, which uses a Tanh activation function to normalize the latent representation within a bounded range.

For the experiments in Section 4.2, we utilized simple convolutional neural networks (CNNs) combined with fully connected layers, ReLU activations, and Tanh activations. In both the PolyMNIST and MVShapeNet datasets, each view is encoded and decoded using separate Variational Autoencoders (VAEs). The architecture of a single VAE's encoder and decoder is summarized in Table 8. The same architecture is applied to each of the five views. The filter sizes ($filter\_tuple$) for PolyMNIST (input shape [3, 28, 28]) and MVShapeNet (input shape [3, 64, 64]) are $(64, 128, 256, 512)$ and $(64, 128, 256, 512, 512)$, respectively.

Table 8: CNN architecture used for each VAE on the PolyMNIST and MVShapeNet datasets.

| Module | Layer Type | Description |
|---|---|---|
| Encoder | Conv2D Layers | 4 layers with increasing filter sizes $filter\_tuple$ and stride of 2. Each followed by ReLU activation. |
| | Flatten | Flatten output to a vector of size 2048. |
| | Linear | Fully connected layer to latent dimension ($d$), followed by ReLU. |
| | Linear (Mu, LogVar) | Separate layers for mean and log variance of latent distribution. |
| Decoder | Linear | Fully connected layer from concatenated latent vector ($d + k$) to output size of 2048. |
| | ConvTranspose2D Layers | 4 layers with decreasing filter sizes $Reverse\_filter\_tuple$, followed by ReLU activations. |
| | ConvTranspose2D (Output) | Final layer with 3 filters, followed by Tanh activation to output RGB image. |

The correspondence between each pair of views is modeled using a simple fully connected network with linear layers and LeakyReLU activations. For all datasets in Section 4.1, we use the architecture $d$ - 128 - 256 - 128 - $d$. For MVShapeNet, we use $d$ - 256 - 512 - 256 - $d$, and for PolyMNIST, we use $d$ - 256 - 1024 - 256 - $d$. The larger network for PolyMNIST reflects the greater differences between its views, requiring more parameters to effectively model the correspondence.

For more implementation details, please refer to the code provided in the supplementary material.

## C.2 MISSING PATTERNS AND MASK GENERATION

We follow standard practices in incomplete multi-view learning, where missing-view masks are generated before training to inform the model of the missing patterns. Following the approach outlined in the public repository by Tang & Liu (2022), for a dataset with $L$ views and a missing rate $\eta$, we randomly select $\eta \times$ len(dataset) samples to be incomplete. For each of these samples, we randomly remove between 1 and $L - 1$ views, ensuring that every incomplete sample retains at least one view while missing at least one. The missing-view masks (e.g., '00101') are generated for the entire dataset before training and stored in a "fingerprint" file for each missing rate $\eta$. When the dataset is loaded, the corresponding masks are applied as long as $\eta$ is specified. This ensures consistent missing patterns across different models, enabling fair and reproducible evaluations.

## C.3 PRE-COMPUTATION OF CYCLIC PERMUTATION INDICES AND BATCH PROCESSING

Given an index set $A = \{1, 2, \ldots, n\}$, Sattolo's Algorithm can efficiently generate a cyclic permutation, as outlined in Algorithm 1. By leveraging stack operations, all possible cyclic permutations of the index set can be precomputed.

For incomplete multi-view datasets with fixed missing patterns, each sample has a distinct mask (e.g., '01101'), indicating the available views $\{2, 3, 5\}$. Precomputing the cyclic permutations for these masks reduces computational overhead during training. Treating missing views as fixed points (e.g., views 1 and 4 in this case) ensures that the permuted index sets maintain the same length (e.g., $[1, \mathbf{2}, \mathbf{3}, 4, \mathbf{5}] \Rightarrow [1, \mathbf{3}, \mathbf{5}, 4, \mathbf{2}]$), simplifying batch operations. These precomputed permutations are also stored in the "fingerprint" file for easy retrieval.

During training, these precomputed permutation indices can be directly applied to arrays containing values for each view, enabling efficient rearrangement of data. For a dataset with $L$ views, cyclic permutations are applied to the latent variables of each sample by permuting elements within each column of an $L \times L$ matrix $Z_0$. As shown in Figure 5, the precomputed cyclic permutations are chosen, structured as an $L \times L$ array and are used to reorder $Z_0$ into $Z_1$. Since these permutations are stored with the dataset, the transformation process can be efficiently executed in batches using indexing, allowing for fast and seamless operations during training.

## C.4 OVERALL TRAINING PIPELINE OF OUR METHOD

The general pipeline of our method is outlined in Algorithm 2. The goal is to learn multiple encoders, decoders, and inter-view correspondences. To implement more easily, we organize the variables into a matrix $Z_0$. The pipeline consists of the following four steps for an $L$-view dataset:

1. **Self-view Encoding and Cross-view Transformation for** $Z_0$: We employ $L$ encoders to obtain $z_v^{(v)} \sim \mathcal{N}(\mu(x_v^{(v)}), \Sigma(x_v^{(v)}))$, $v \in \{1, \ldots, L\}$. Then, using all inter-view correspondences $\{f_{lv}\}$, we compute $z_v^{(l)} \sim \mathcal{N}(f_{lv} \circ \mu(x_v^{(v)}), f_{lv} \circ \Sigma(x_v^{(v)}))$, for $l \neq v$, where $l, v \in \{1, \ldots, L\}$. These values are organized into a matrix $Z_0 = \left[z_v^{(l)}\right]_{L \times L}$, where the row index denotes the subscript and the column index denotes the superscript.

2. **Cyclic Permutation for** $Z_1$: We apply $L$ pre-generated cyclic permutations $\{\sigma_l\}_{l=1}^L$, provided with the dataset, to form an permuted index array of size $L \times L$. This array is used to transform $Z_0$ into $Z_1 = \left[z_{\sigma_l(v)}^{(l)}\right]_{L \times L}$.

3. **Fusion for** $\omega$ **with Complete-view Partition**: For both $Z_0$ and $Z_1$, we fuse each row by computing the geometric mean of the $k$-dimensional marginal distributions, resulting in $\Omega_0 = \{\omega_l^0\}_{l=1}^L$ and $\Omega_1 = \{\omega_l\}_{l=1}^L$. We also extract the diagonal elements from $Z_0$ and $Z_1$, corresponding to $\{z_l^{(l)}\}_{l=1}^L$ and $\{z_{\sigma_l(l)}^{(l)}\}_{l=1}^L$, respectively, where $\sigma_l(l) \neq l$.

4. **Decoding with** $\omega$ **and** $z$: We use $z$ to reconstruct the view indicated by its superscript, combining it with a consensus variable $\omega$. Specifically, we apply $L$ decoders to reconstruct the views as follows: $\hat{x}^{(l)} = \text{Decoder}([\omega_l^0, z_l^{(l)}])$ and $\tilde{x}^{(l)} = \text{Decoder}([\omega_l, z_{\sigma_l(l)}^{(l)}])$.

5. **Masked ELBO Update**: We apply masks to ignore terms related to missing views and update the model using the ELBO objective (Eq. 4).

---

**Algorithm 2** Multi-View Permutation of VAEs for Incomplete Data

---

**Input:** Incomplete Multi-View dataset $\{\mathbb{X}_i\}_{i=1}^n$, latent variable dimension $d$, and shared feature dimension $k$ $(k \leq d)$.
**Output:** MVAEs with inter-view correspondences.
Initialize parameters $\{\phi_v\}$, $\{\theta_v\}$, and $\{\alpha_{lv}\}$, where $v, l \in [L]$.
 1: **while** maximum epochs not reached **do**
 2:     **for** $v$ in available views **do**
 3:         Generate $(\mu_v^{(v)}, \Sigma_v^{(v)}) = \text{Encoder}(x^{(v)})$ through $v$-th encoder;
 4:         Compute $(\mu_v^{(l)}, \Sigma_v^{(l)})$ using $f_{lv}$ for $l \neq v$, $l \in \{1, 2, \ldots, L\}$;
 5:     **end for**
 6:     For derived $\mathbf{Z}_0$, apply cyclic permutations to obtain $\mathbf{Z}_1$;
 7:     **for** $n$ in available views **do**
 8:         Calculate geometric mean $(\alpha_n, \Lambda_n)$ from $\mathcal{C}_n$ and sample $\omega_n$;
 9:         Sample $z_*^{(n)}$ from single-view cell $\mathcal{S}_n$;
10:         Generate $\hat{x}^{(n)} = \text{Decoder}(\omega_n, z_*^{(n)})$ through $n$-th decoder;
11:     **end for**
12:     Update $\{\phi_v\}$, $\{\theta_v\}$, and $\{\alpha_{lv}\}$ by maximizing the ELBO (Eq. 4);
13: **end while**

---

Table 9: **Overview of comparative methods for incomplete multi-view learning.** This table summarizes the key modeling details of the methods used for comparison.

| Method | Year | Venue | #View | Description |
|--------|------|-------|-------|-------------|
| DCCA | 2013 | ICML | Two | Baseline, CCA-based method |
| DCCAE | 2015 | ICML | Two | Baseline, CCA-based method |
| DIMVC | 2022 | AAAI | Mutiple | High-dimensional nonlinear mapping, EM |
| DSIMVC | 2022 | ICML | Mutiple | Bi-level opimization |
| Completer | 2023 | CVPR/TPAMI | Two | Information theory, contrastive learning |
| CPSPAN | 2023 | CVPR | Mutiple | Prototype alignment |
| ICMVC | 2024 | AAAI | Two | GNN, high-confidence guiding |
| DVIMC | 2024 | AAAI | Mutiple | GMM+VAEs |

## C.5 INFERENCE OF MISSING VIEWS WITH AVAILABLE VIEWS

During inference, given an incomplete input sample $\{x^{(v)}\}_{\mathcal{I}}$, we first derive the latent variable set $\{z_v^{(l)}\}_{(v,l) \in \mathcal{I} \times [L]}$. For each view $l \in \{1, \ldots, L\}$, we compute an averaged latent variable $\bar{z}^{(l)}$ by taking the geometric mean of the distributions from the available views, $\bar{z}^{(l)} = \text{Geometric Average}(\{z_v^{(l)}\}_{v \in \mathcal{I}})$. This process provides a complete set of $L$ latent variables $\{\bar{z}^{(l)}\}_{l=1}^L$.

Next, we compute a consensus variable $\omega$ from this complete set by taking the geometric mean of their marginal distributions over the first $k$ dimensions. Finally, we concatenate $\omega$ with each $\bar{z}^{(l)}$ and use the $l$-th decoder to reconstruct $x^{(l)}$, allowing us to infer all views.

## D COMPARATIVE METHODS

### D.1 INCOMPLETE MULTI-VIEW LEARNING METHODS IN SECTION 4.1

For the methods listed in Table 9, we follow their original implementations as provided in the respective public repositories. For methods that are limited to handling only two views, we train them on all possible combinations of two views and report the best performance for each dataset.

Table 10: **Comparative methods for different multi-modal VAEs.** This table outlines key details of various multi-modal VAE approaches used for comparison, including the latent variable dimension, joint posterior fusion strategy, and prior setting.

| Method | Year | Venue | Dimension | Joint Posterior | Prior and KL divergence term |
|--------|------|-------|-----------|-----------------|------------------------------|
| mVAE | 2018 | NeurIPS | 512 | PoE | Spherical Gaussian: $\mathcal{N}(0,1)$ |
| mmVAE | 2019 | NeurIPS | 512 | MoE | Laplace distritbution |
| mmJSD | 2021 | NeurIPS | 512 | PoE/MoE | JS divergence with Geometric Mixture |
| MoPoE | 2021 | ICLR | 512 | MoPoE | Spherical Gaussian: $\mathcal{N}(0,1)$ |
| MVTCAE | 2021 | NeurIPS | 512 | PoE | Spherical Gaussian: $\mathcal{N}(0,1)$, Total Correlation objective resulting in forward KL |
| mmVAE+ | 2023 | ICLR | 96/256 | MoE | Laplace distritbution |

## D.2 Multimodal VAEs in Section 4.2

Table 10 summarizes the key modeling details of the various multi-modal VAE approaches used in our comparisons, including latent space dimensions, joint posterior fusion strategies, and prior settings. We adhere to the implementations provided in their respective public repositories. For the first five models, we set the latent dimension to 512 for both the PolyMNIST and MVShapeNet datasets. For mmVAE+, we use latent dimensions of 96 for PolyMNIST and 256 for MVShapeNet, matching the settings used in our method. Following its original implementation, 50% of the latent dimension is allocated to the shared subspace, with the remaining 50% designated for view-specific subspaces. All methods were trained for 300 epochs across all datasets.

## E Dataset Information and Construction Details

Table 1 provides an overview of the statistics for all datasets used in this study, all of which are publicly available through their respective repositories and on the Huggingface website. For PolyMNIST (Sutter et al., 2021), we present sample images in Figure 14.

The MVShapeNet dataset was constructed from the ShapeNet dataset (Chang et al., 2015) using the rendering tool developed by Stanford[2]. We selected five categories: table, chair, car, airplane, and rifle, with a total of 25155 samples. The number of samples for each category is 8445, 6778, 3514, 4045, and 2373, respectively.

To convert the 3D point cloud data into multiple 2D views, we employed a directional lighting setup. The primary light source was configured as a sun light with shadows disabled, a specular factor of 1.0, and an energy level of 10.0. To ensure consistent lighting across surfaces not directly illuminated by the primary source, a secondary sun light was added with a low energy level of 0.015, positioned 180° relative to the primary light. Both light sources had shadows disabled to maintain uniform illumination. The camera was placed at coordinates (0, 1, 0.6), with a focal length of 35mm and a sensor width of 32mm. To capture multiple views, the camera was constrained to track an empty object at the origin, which was rotated in 8° increments (360°/45 steps) around the Z-axis. We selected five views, each taken from viewpoints spaced 45 degrees apart around the front of the object. Example images from the MVShapeNet dataset are shown in Figure 15.

---

[2]https://github.com/panmari/stanford-shapenet-renderer

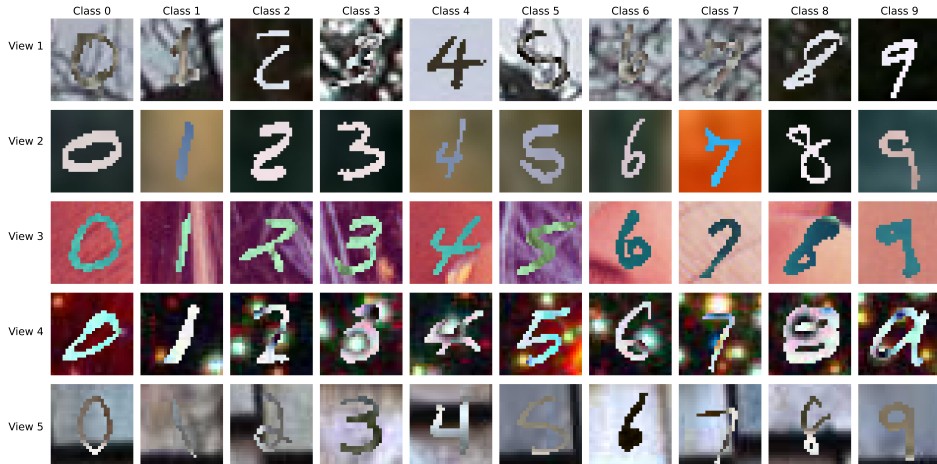

Figure 14: Visualization of randomly selected MNIST samples across 10 digit classes (0-9) and 5 modalities (m0-m4). Each column represents a digit class, and each row shows a different modality.

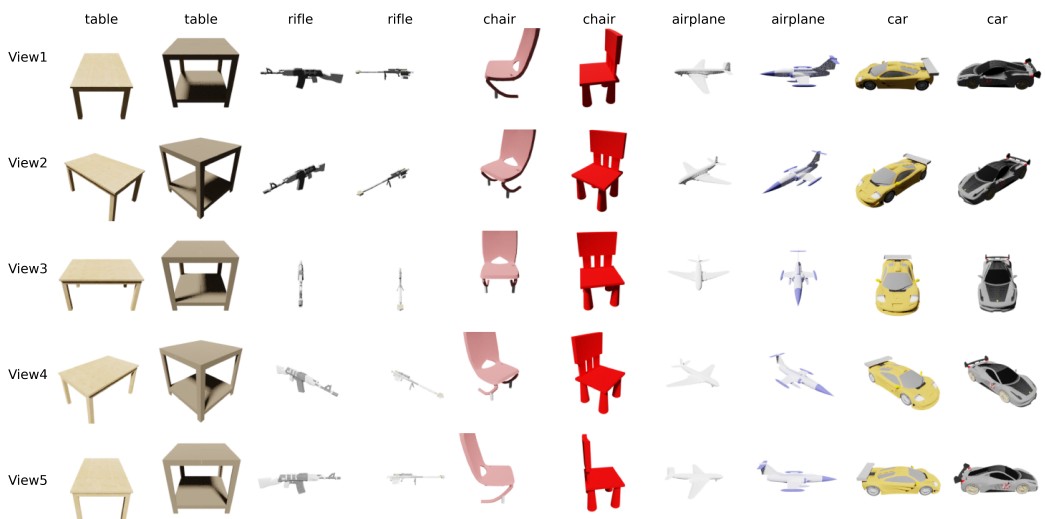

Figure 15: Visualization of randomly selected MVShapeNet samples from 5 object categories. Each row shows 5 different views (View1-View5) of two randomly selected samples from each category.

