# OpenReview forum: "Deep Incomplete Multi-view Learning via Cyclic Permutation of VAEs"
_ICLR.cc/2025/Conference — ICLR 2025 Poster_

### Official Review · Reviewer_yAd5 · 2024-10-29

**Soundness:** 3
**Presentation:** 2
**Contribution:** 3
**Rating:** 6
**Confidence:** 4

**Summary:**

This paper introduces the Multi-View Permutation of Variational Autoencoders (MVP) for incomplete multi-view representation learning. By leveraging cyclic permutations of posteriors, MVP enhances inter-view consistency and infers missing views effectively. Key elements include partitioning variables for view invariance, deriving an Evidence Lower Bound (ELBO) for optimization, and implementing an informational prior through cyclic permutation to align distributions across views. Experimental validation across multiple datasets and missing data scenarios shows MVP’s superiority over existing methods in clustering and generation tasks, particularly under high missing rates.

**Strengths:**

- MVP introduces a cyclic permutation approach to VAEs, leveraging view-invariant transformations in the latent space to address the challenges posed by incomplete multi-view data. This method successfully captures inter-view relationships by modeling correspondences and creating a more robust latent space, which significantly enhances the ability to infer missing views and aggregate information effectively.
- The paper provides rigorous theoretical analysis and proofs.
-  The extensive experiments, particularly in scenarios with high missing data ratios, provides compelling evidence of MVP’s ability to handle incomplete data effectively. The authors not only validate the approach across different missing ratios but also include quantitative metrics and visualizations to showcase MVP’s advantage over competing models.

**Weaknesses:**

##### Major

- "Even if we reorder the variables within each column of Z0, as demonstrated in the transition from Z0 to Z1in Figure 1, the underlying semantic information remains invariant"? The explanation of “underlying semantic information” remains unclear. A detailed clarification on what specific semantic information remains invariant during these transformations would strengthen the paper.
- MVP’s performance could be sensitive to cyclic permutation settings and regularization parameters, as they directly influence view consistency and the similarity measure. A deeper sensitivity analysis of these parameters would provide insight into MVP’s stability and adaptability.
- The cyclic permutation technique appears conceptually similar to disturbance and re-alignment strategies as proposed in *Partially View-Aligned Clustering* (NeurIPS ’20). A discussion of distinctions and similarities between MVP and this approach would benefit readers.
- Network architectures used for MVP are not fully detailed in the paper, limiting reproducibility and making it challenging for readers to understand the baseline structures supporting MVP’s performance.

##### Minor

- Highlighting the Single-view Partition and Complete-view Partition in Figure 1 would improve clarity.
- The transition from Z_0 to Z_1 in Figure 1 is difficult to follow; linking this to Section A.2 might assist readers.
- It is not easy to distinguish the views with the samples in Fig.1b. It would be better to use different number of views and samples.
- The reference to “The second term” in Line 303 would be clearer if accompanied by Equation 1.

**Questions:**

See Major problems in Weaknesses

---

> ### Author Response · Authors · 2024-11-21
> **Official Comment to Reviewer yAd5**
>
> We deeply appreciate your thoughtful and constructive feedback, which has been invaluable in refining and improving our manuscript. Below, we provide detailed responses to your insightful comments:
> ## Q1. Clarification on "Underlying Semantic Information"
>
> Thank you for your careful review of both the main text and the appendix. We appreciate your insightful observation and acknowledge that the term **"underlying semantic information"** was indeed too vague and lacked precision. Specifically, we intended to convey that the **encoded information of the latent variables in matrix $Z_0$ remains invariant under specific permutations**. To address this, we have revised the relevant text (**lines 150–154, highlighted in blue**) to provide better clarity and context. The updated text reads as follows:
>
> "The core idea is that variables with the same superscript $l$ encode similar information about the $l$-th view, regardless of whether they are directly encoded or transformed. Thus, even if the columns of $Z_0$ are reordered, as illustrated in the transition from $Z_0$ to $Z_1$ in Figure 1, each column continues to represent the same view, and the information encoded by elements at corresponding positions remains invariant."
>
> This revision clarifies that the latent variables consistently represent view-specific information even after reordering, enhancing the clarity of our approach.
> ## Q2. Sensitivity to Regularization Parameters and Cyclic Permutation Setting
>
> We appreciate your inquiry regarding the sensitivity of our method to regularization parameters and cyclic permutation settings.
>
> As detailed in **Appendix B.4**, we analyzed the sensitivity of the KL divergence regularization parameters for $z$ and $\omega$, selecting 5 and 2.5, respectively. To further investigate, we performed a finer-grained analysis and visualized the results as a 3D surface plot (**Figure 13**). **The relatively flat surface indicates overall robustness**. Importantly, the KL term for $z$ benefits from a larger value, as it is critical for establishing inter-view correspondences, while the KL term for $\omega$ serves as a smaller auxiliary regularizer to enforce consistency.
>
> Regarding the potential sensitivity to cyclic permutations, we understand the reviewer’s concern about the randomness introduced by different cyclic permutations (e.g., 1234 cyclically permuted to 2341 or 3421). **Actually, we pre-compute all possible cyclic permutations for the observed view indices and uniformly sample from them during training**. This ensures that, over sufficient training iterations, all permutations are equally represented. Furthermore, the goal of cyclic permutations is consistent—to encourage all distributions within the same set ($\mathcal{S}_i$ in our paper) to become similar. Our ablation study (**Table 5**) includes **a comparison between cyclic and random permutations, indeed validating the crucial role of cyclic permutations in establishing a strong similarity measure**. Finally, all experiments are run with multiple random seeds, reporting mean and standard deviation, demonstrating robustness to stochastic effects from permutation sampling.
>
> In conclusion, the cyclic permutations ensure consistency, and the regularization parameters are robust under proper tuning, as validated by our analysis. We sincerely thank you again for your thoughtful review.

---

> > ### Author Response · Authors · 2024-11-21
> > **(2) Official Comment to Reviewer yAd5**
> >
> > ## Q3. Comparison with Partially View-Aligned Clustering
> >
> > Thank you for suggesting a comparison between our MVP method and the disturbance and re-alignment strategies in **Partially View-Aligned Clustering (PVC, NeurIPS '20)**. MVP and PVC address two distinct challenges in multi-view learning: the **partially data-missing problem (PDP)** for MVP and the **partially view-aligned problem (PVP)** for PVC. Both methods leverage inherent inter-view consistency in multi-view data, but despite some overlapping terminology, our modeling approaches are fundamentally different.
> >
> > MVP establishes **continuous correspondences** between views through explicit mapping functions to **infer missing information and effectively complete the latent space**. We then apply **cyclic permutation** to reorder latent variables while preserving semantic consistency, resulting in a permutation-invariant structure that allows us to derive a valid Evidence Lower Bound (ELBO) for optimization.
> >
> > In contrast, PVC formulates the **alignment problem** as an **integer linear programming (ILP)** task, solved using Dykstra’s projection—a differentiable approximation of the Hungarian algorithm. The goal in PVC is to optimize a **binary permutation matrix** that indicates whether two views are aligned (i.e., whether they originate from the same sample). Unlike MVP, which establishes continuous transformations to link views, PVC focuses solely on aligning discrete matching parts without inferring deeper latent relationships.
> >
> > In summary, MVP effectively mitigates information bias arising from complex missing patterns, providing a robust solution for incomplete multi-view data. Due to the distinct nature of the problems we address, PVC was not included in the related work section because of space limitations, though **we acknowledge its conceptual relevance and have cited it appropriately in our introduction and methods (Line 63 and Line 136-138)**.
> > ## Q4. Detailed Network Architectures
> >
> > Thank you for your thoughtful feedback regarding the need for more details on the network architectures used in MVP to enhance reproducibility.
> >
> > To address this, we have made the following revisions:
> > 1. **Appendix Addition**: We have added detailed descriptions of the network architectures in the appendix (see **Appendix C.1**). Specifically, we clarify that we used simple fully connected neural networks and convolutional neural networks (CNNs) similar to those used in previous studies.
> > 2. **Supplementary Code**: The code for MVP, including all baseline network structures, is available in the supplementary material and on an anonymous GitHub repository. This will allow you and other readers to fully understand and reproduce the results presented in the paper. We will continue to improve and refine the code to make it more user-friendly and efficient in the future.
> >
> > We are committed to continuously improving the clarity and usability of our codebase to support future research.
> >
> > ## Q5. Minor writing and figure issue
> >
> > We sincerely thank you for your thoughtful and detailed suggestions, which have significantly enhanced the clarity of our paper.
> > 1. **Figure 1 Improvements**
> >     - We have updated Figure 1 to explicitly highlight the Single-view and Complete-view Partitions and clarified the transition from Z_0​ to Z_1​ with an improved caption and a direct reference to Section A.2.
> > 2. **Reference to "The second term" in Line 303**
> >     - We have revised Line 303 to explicitly reference Equation (1), ensuring greater clarity and readability.
> >
> > Once again, we sincerely thank you for your thoughtful feedback, which has been instrumental in strengthening our work. We hope the revised manuscript addresses your concerns comprehensively and effectively.

---

> ### Author Response · Authors · 2024-12-02
> **Anticipating Your Response**
>
> Dear Reviewer yAd5,
>
> Thank you sincerely for your efforts in reviewing our paper. We have carefully addressed your concerns with corresponding responses and results. We would appreciate your feedback to confirm if our clarifications resolve your concerns. Please feel free to let us know if any part remains unclear.
>
> Best regards,
>
> Authors

---

> > ### Comment · Reviewer_yAd5 · 2024-12-03
> >
> > Thanks for the response, I'm happy to keep my rating.

---

### Official Review · Reviewer_7w7m · 2024-10-31

**Soundness:** 3
**Presentation:** 3
**Contribution:** 3
**Rating:** 6
**Confidence:** 5

**Summary:**

In this paper, the authors design a multi-view permutation of variational auto-ecoders for incomplete multi-view clustering, which termed MVP. The variational auto-encoder is used for extracting the invariant features.  Randomly reoder variables are used for cross-view generation.

**Strengths:**

1. Extensive experiments have demonstrated the effectiveness of the methodology designed in the paper.

2. The paper is well-organized.

3. The motivation is clearly described.

**Weaknesses:**

1. The use of VAE for invariant feature learning has been extensively studied [1,2,3, 4]. Please analyze the differences.

2. The paper does not provide the code, yet the results in Table 2 show significant performance of the proposed method. The authenticity and fairness of the experiments are questionable. Therefore, please release the code during the rebuttal process, as this will be a key criterion in my evaluation.

3. The construction details of the PolyMNIST and MVShapeNet datasets in line 406 are unclear. Is using new datasets to test previous methods fair?

4. How can the impact of randomness introduced by random padding, as mentioned in line 20, be mitigated?


5. The novelty needs to be further clarified. The use of Variational Autoencoders and ELBO has been widely studied in the IMVC field.

[1] Xu G, Wen J, Liu C, et al. Deep Variational Incomplete Multi-View Clustering: Exploring Shared Clustering Structures[C]//Proceedings of the AAAI Conference on Artificial Intelligence. 2024, 38(14): 16147-16155.

[2] Cai H, Huang W, Yang S, et al. Realize Generative Yet Complete Latent Representation for Incomplete Multi-View Learning[J]. IEEE Transactions on Pattern Analysis and Machine Intelligence, 2023.

[3]Chen M, Huang H, Li Q. Towards Robust Uncertainty-Aware Incomplete Multi-View Classification[J]. arXiv preprint arXiv:2409.06270, 2024.

[4] Xu J, Ren Y, Tang H, et al. Multi-VAE: Learning disentangled view-common and view-peculiar visual representations for multi-view clustering[C]//Proceedings of the IEEE/CVF international conference on computer vision. 2021: 9234-9243.

**Questions:**

Please see weakness.

---

> ### Author Response · Authors · 2024-11-21
> **Official Comment to Reviewer 7w7m**
>
> We sincerely appreciate your thorough and constructive feedback, which has been instrumental in improving the clarity and depth of our work. Below, we address your insightful comments in detail:
> ## Q1. Comparison of Invariant Feature Learning: Other Works vs. Our Novelty
>
> Thank you for the opportunity to highlight the novelty of our work and its distinctions from prior methods.
>
> _The essence of multi-view data lies in capturing both shared and unique information across diverse modalities or perspectives._ Many methods have been proposed to leverage this inherent property, but **prior works have not fully explored the information potential of multi-view data within the VAE framework, particularly under incomplete scenarios.** For instance:
> - **DIMVC [1]**: Enforces **strict consistency** between views to retain only shared information, reducing reliance on missing views. While this improves clustering performance, it has **not been validated for generative tasks**.
> - **Multi-VAE [3]**: Models view-common information using **discrete distributions** to achieve **strict disentanglement, minimizing the influence of view-specific details on clustering**. However, it requires concatenating representations from all views, making it **unsuitable for incomplete scenarios**.
> - **APLN [4]**: Addresses **uncertainty and conflicts in multi-view classification** caused by imputation, focusing on resolving conflicting opinions. This method differs fundamentally from ours, relying on label supervision and a multi-stage iterative strategy tailored to classification tasks.
> - **CMVAE [2]**: Assumes **invariant linear relationships between views** in the latent space. This strict assumption may limit the preservation of view-specific details, as seen in blurred PolyMNIST backgrounds. Its GMM-based latent space, designed to capture class information, requires predefined modes and is less effective at learning inter-view relationships.
>
> We would like to emphasize that, unlike previous works, our method **fully leverages the invariant relationships between views** through two key innovations:
> 1. **Embedding Invariant Relationships into the Latent Space**:
>     We establish explicit correspondences between views to flexibly capture their invariant relationships. By leveraging set permutations and partitions, we directly embed these relationships into the variational inference framework, ensuring that the latent space inherently reflects this invariance.
> 2. **Latent Space Regularization with Inter-View Relationships**:
>     We introduce a prior based on inter-view relationships to self-supervise latent variable learning, enhancing efficiency and creating a synergistic interplay between inter-view correspondences and the latent space. This interplay makes our method more flexible than [2].
> Furthermore, our method does not require strict alignment between views. Instead, we adopt a **soft alignment** approach, where views only need to be consistent after transformation. This flexibility enables the latent variables to **preserve both view-common and view-specific information** while facilitating the **imputation of missing views through correspondences.** Our framework effectively addresses **information imbalance and bias caused by incomplete fusion**, a critical challenge that has not been well-addressed within the variational framework for incomplete multi-view learning. Finally, our representation has been thoroughly validated through clustering and generation tasks, demonstrating its robustness and versatility.
>
> We hope this clarifies the unique contributions of our work and its advancements over existing methods.
>
> ## Q2. Publication of Code
>
> We understand the importance of reproducibility and have taken several steps to ensure transparency. The code has been included in the supplementary material and is also available in an anonymous GitHub repository: https://anonymous.4open.science/r/MVP-F154/ . This repository includes expanded technical details, such as implementations for cyclic and batch permutations, to facilitate reproduction.
>
> To further ensure accessibility, we will release complete implementations for all publicly available datasets used in our paper. We are also committed to refining and optimizing the code for enhanced usability.
>
> We hope this demonstrates our dedication to transparency and reproducibility.

---

> > ### Author Response · Authors · 2024-11-21
> > **(2) Official Comment to Reviewer 7w7m**
> >
> > ## Q3. Construction details of the PolyMNIST and MVShapeNet
> >
> > We appreciate the opportunity to clarify the construction of these datasets and their use in evaluating prior methods.
> >
> > **PolyMNIST**: This dataset, first introduced by **MoPoE** (ICLR '21 [5]), is widely used in multi-view learning research (all comparative methods in Section 4.2). We followed the original protocols and generation scripts provided by the authors to ensure consistency and fairness in our evaluations.
> >
> > **MVShapeNet**: We fully understand the concerns about fairness when using a newly constructed dataset for comparisons. **MVShapeNet** was inspired by feedback provided during the OpenReview stage for **MVTCAE** (NeurIPS '21, [6]), where reviewers suggested expanding evaluations beyond PolyMNIST to include datasets that reflect broader physical-world scenarios for "views." In response, MVTCAE incorporated the **Multi-PIE** dataset (250 subjects, 3 views) featuring multi-view human faces into their evaluations. Drawing from this idea, we developed **MVShapeNet** to complement existing benchmarks by providing a dataset with **five standardized viewpoints and well-defined categories**. The construction process for MVShapeNet is detailed in **Appendix E**, followed standard practices in multi-view learning, using the **Stanford rendering tool**. To ensure transparency and reproducibility, we will release all data and code associated with MVShapeNet.
> >
> > Beyond these two datasets, we conducted experiments on additional commonly used multi-view learning datasets (**Table 1**) to provide comprehensive validation. We hope this clarifies the rigor and fairness of our evaluations.
> >
> > [5] Sutter T M, Daunhawer I, Vogt J E. Generalized Multimodal ELBO[C]//International Conference on Learning Representations (ICLR 2021). OpenReview, 2021.
> >
> > [6] Hwang H J, Kim G H, Hong S, et al. Multi-view representation learning via total correlation objective[J]. Advances in Neural Information Processing Systems, 2021, 34: 12194-12207.
> >
> > ## Q4. Clarification on Randomness and Its Impact
> >
> > We understand that you may be referring to the **randomly reordering variables** mentioned in **line 20** and are concerned about the potential impact of randomness introduced by permutation.
> >
> > In our method, randomness, particularly in the form of **cyclic permutations**, plays a key role in improving cross-view consistency. Random reordering is a common technique in multi-view learning. For example, Huang et al. (NeurIPS '20) used permutations in multi-view data to learn a matching matrix, framing it as a matching problem. Unlike standard random permutations, cyclic permutations are carefully designed to enforce a **permutation-invariant latent space structure** while preserving inter-view relationships.
> >
> > In our derived ELBO, randomness plays a crucial role. The **reconstruction term** benefits by enabling the model to jointly optimize **within-view reconstruction** and **cross-view generation**, while the two **KL regularization terms** are strengthened by setting the prior as a cyclic permutation of the posterior.
> >
> > Experimentally, we conducted an ablation study **comparing cyclic permutations with standard random permutations**, and the results demonstrated the superiority of cyclic permutations in improving model perform
> > ance. Furthermore, latent space visualizations during training confirmed the **stability** of our method, with individual loss terms converging as expected. The **robustness** of our approach is further supported by clustering and generation results, where our model consistently outperforms baselines across multiple metrics.
> >
> > We hope this explanation addresses your concerns and highlights the importance of randomness within our framework.
> >
> > ---
> >
> > Once again, we are grateful for your thoughtful and detailed feedback. Your comments have been invaluable in strengthening our work, and we hope the revised manuscript addresses your concerns effectively.

---

> > > ### Author Response · Authors · 2024-12-02
> > > **Anticipating Your Response**
> > >
> > > Dear Reviewer 7w7m,
> > >
> > > Thank you sincerely for your efforts in reviewing our paper. We have carefully addressed your concerns with corresponding responses and results. We would appreciate your feedback to confirm if our clarifications resolve your concerns. Please feel free to let us know if any part remains unclear.
> > >
> > > Best regards,
> > >
> > > Authors

---

### Official Review · Reviewer_dPkn · 2024-11-03

**Soundness:** 2
**Presentation:** 2
**Contribution:** 2
**Rating:** 5
**Confidence:** 4

**Summary:**

This paper introduces the MVP framework named of Multi-View Permutation of Variational Auto-Encoders, which focuses on a practical problem for incomplete multi-view learning. MVP can use MVAEs to establish view relationships in the latent space, thereby aggregating more comprehensive information while inferring missing views. Compared with the existing methods, the authors arrange and partition the variables and use a circular permutation approach to transform regularization into a measure of distribution similarity, thereby enhancing the consistency between different perspectives. The experimental results on several real-world datasets demonstrate the effectiveness of the proposed model.

**Strengths:**

1. This paper is well-motivated since missing is a common and significant problem in real-world multi-view learning. The modelling of the missing problem in this paper is very well designed and the method of applying permutations and segmentations in the latent space is very novel.
2. The paper provides a well-structured overview that is easy for the reader to understand. In addition, implementation details of the selected technology are presented in detail.
3. The paper provides comprehensive experimental results to validate the effectiveness of the proposed method. The model is tested on seven different datasets and compared with several SOTA methods to demonstrate its usefulness and robustness

**Weaknesses:**

1. The complexity of the proposed method is not adequately discussed. It would be helpful to compare the computation cost of the proposed method to the baselines.
2. This paper assumes that the first k dimensions of z capture information common to all views, so how is it set on different datasets?
3. This paper only shows the results generated on the PolyMNIST and MVShapeNet datasets, where the views are all RGB type. How are the results generated on datasets with other types of views, such as the CUB Dataset?
4. For the experimental results, why are the experimental results you provided lower than the original papers? For example, DVIMC on the Scene 15 dataset with missing rates η=0.1 is lower than the published papers. In addition, how were the experimental results of Completer obtained? The original Completer is proposed for two view data which cannot be applied on the datasets you exploited in the paper directly.
5. The convergence analysis can be added in the experiment, which can be adopted to better the loss function.

**Questions:**

None

---

> ### Author Response · Authors · 2024-11-21
> **Official Comment to Reviewer dPkn**
>
> We are deeply grateful for your thoughtful and constructive feedback, which has greatly contributed to enhancing the clarity and rigor of our work. Below, we provide detailed responses to each of your comments:
> ## Q1. Computational Complexity of MVP
> To address your concern regarding the computational complexity, we have conducted a detailed comparison of our method and the baseline methods. The table below highlights the computational cost of encoding, transformation, and decoding across different approaches:
>
> |                    | MVAE   | **MMVAE** | **mmJSD** | **MoPoE** | **MVTCAE** | **Ours**    |
> | ------------------ | ------ | --------- | --------- | --------- | ---------- | ----------- |
> | **Encoding**       | $O(V)$ | $O(V)$    | $O(V)$    | $O(2^V)$  | $O(V)$     | $O(V)$      |
> | **Transformation** | /      | /         | /         | /         | /          | $O(V(V-1))$ |
> | **Decoding**       | $O(V)$ | $O(KV^2)$ | $O(V)$    | $O(V)$    | $O(V)$     | $O(V)$      |
>
> Here, $V$ denotes the number of views, and $K$ is the number of latent variable samples per view for optimizing the IWAE bound. While our method introduces additional computational cost during the transformation step due to explicit channel construction in the latent space, these operations are performed using a lightweight MLP in a low-dimensional space, keeping the overall complexity manageable.
>
> We hope this analysis clarifies the trade-offs of our approach and provides a clear understanding of its computational efficiency.
>
> ## Q2. Latent Dimension Selection for Different Datasets
>
> Thank you for your question regarding how we select the first $k$ dimensions of $z$ to encode shared information across different datasets.
>
> First, to prevent any potential misunderstanding, we would like to clarify that our approach is distinct from feature or latent variable disentanglement methods [1], which aim to capture shared information strictly for interpretability. In our method, $k$ is used to derive an average representation $\omega$ from the latent variables $z$ of different views, providing additional regularization while allowing flexibility by choosing $k \leq d$.
>
> Second, we conducted ablation studies, detailed in **Appendix B.4**, to illustrate the flexibility of our approach in selecting $k$ based on task requirements. Below is a summary of our findings:
> - **Clustering Tasks**: For clustering, strong inter-view consistency is crucial to effectively capture shared information and reveal underlying categories. Thus, we set $k = d$, using all latent dimensions. Our ablation results show that the optimal choice of $k$ depends on dataset characteristics (e.g., view dimensions, number of classes, **Table 7**). Smaller datasets benefit from careful selection of latent dimensions, while larger datasets exhibit robust performance across different values of $d$, reducing the need for fine-tuning. This Parameter Sensitivity Analysis aligns with established practices [2, 3].
> - **Generation Tasks**: For generation tasks, where retaining unique features from each view is important, we tested different ratios of $k/d$ (25%, 50%, 75%, 100%) and found robust results for both PolyMNIST and MVShapeNet, given their larger size (**Table 8**). Therefore, we chose different ratios based on dataset characteristics: **50%** for PolyMNIST, which has higher view variability, and **75%** for MVShapeNet, where views are more consistent.
>
> We hope this explanation clarifies our rationale and demonstrates the adaptability of our method.
>
> [1] Chen R T Q, Li X, Grosse R B, et al. Isolating sources of disentanglement in variational autoencoders[J]. Advances in neural information processing systems, 2018, 31.
> [2] Zhang C, Cui Y, Han Z, et al. Deep partial multi-view learning[J]. IEEE transactions on pattern analysis and machine intelligence, 2020, 44(5): 2402-2415.
> [3] Cai H, Huang W, Yang S, et al. Realize Generative Yet Complete Latent Representation for Incomplete Multi-View Learning[J]. IEEE Transactions on Pattern Analysis and Machine Intelligence, 2023.

---

> > ### Author Response · Authors · 2024-11-21
> > **(2) Official Comment to Reviewer dPkn**
> >
> > ## Q3. Results on CUB Dataset which consists of both image and texts
> >
> > Your question about our evaluation on the CUB dataset prompted us to expand on our methodology and results.
> >
> > Firstly, we clarify that **Section 4.1 of our paper uses the simplified two-view version of the CUB dataset** as introduced by [4,5]. In this version, pre-trained networks are used to extract image and text embeddings (reduced to 1024 and 300 dimensions, respectively). This processed version has been widely adopted in multi-view clustering and classification tasks. Our clustering experiments also rely on this version, which is not suitable for demonstrating generative performance due to its pre-processed nature.
> >
> > **To showcase the generation results of our method on datasets with non-RGB views, we performed additional experiments on the raw-text version of the CUB dataset** used by MMVAE+ (ICLR '23, [6]). This version consists of paired bird images and text descriptions (padding to same length), with 88,550 training and 29,330 testing samples. Following MMVAE+, we adopted their network architecture and latent dimension (64 dimensions). The **visual results are provided in Appendix B.2.**
> >
> > The raw-text CUB dataset poses significant challenges due to its complexity:
> > - It contains 200 bird species with diverse image-text pairs. While the paired descriptions align with images, they are often **coarse-grained** and include **conflicting or uncertain details**.
> > - Text descriptions primarily focus on broad characteristics like color or structure, while **images include many fine-grained visual details**.
> >
> > In the **incomplete setting**, we tested our method by generating images using only textual descriptions. **The results (Figure 10) demonstrate basic semantic alignments in attributes such as colors (e.g., black, white, brown) and structures (e.g., belly, beak, wings).** The blurry backgrounds and contours, consistent with MMVAE+ observations (Page 23-24 of [6]), stem from the inherent limitations of single-step VAE-based generation. Recent works, such as D-CMVAE [7], further refine VAE-generated results using diffusion models for clearer outputs, but such enhancements fall outside the scope of our current study.
> >
> > Regarding quantitative evaluation, traditional metrics with respect to image diversity or reconstruction scores fail to effectively measure **cross-modal consistency**. MMVAE+ proposed a coherence metric for CUB dataset to evaluate caption-to-image generation by analyzing HSV color distributions in generated images and comparing them to textual color descriptions (Page 17-18 of [6]). While meaningful, this metric does not capture high-level consistencies, such as specific bird attributes (e.g., crown, beak, wings) or environmental contexts (e.g., sky, water, trees). Future work could explore **fine-grained consistency metrics tailored to specific multi-modal datasets** or leverage **large vision-language models for automated evaluation** [8]. **We leave this as an avenue for future work in the journal version of this paper.**
> >
> > Thank you again for this valuable question, which helped us reflect more deeply on the generalizability and evaluation of our method.
> >
> > [4] Netzer Y, Wang T, Coates A, et al. Reading digits in natural images with unsupervised feature learning[C]//NIPS workshop on deep learning and unsupervised feature learning. 2011, 2011(2): 4.
> >
> > [5] Shi Y, Paige B, Torr P. Variational mixture-of-experts autoencoders for multi-modal deep generative models[J]. Advances in neural information processing systems, 2019, 32.
> >
> > [6] Palumbo E, Daunhawer I, Vogt J E. MMVAE+: Enhancing the generative quality of multimodal VAEs without compromises[C]//The Eleventh International Conference on Learning Representations. OpenReview, 2023.
> >
> > [7] Palumbo E, Manduchi L, Laguna S, et al. Deep Generative Clustering with Multimodal Diffusion Variational Autoencoders[C]//The Twelfth International Conference on Learning Representations. 2024.
> >
> > [8] Lin Z, Pathak D, Li B, et al. Evaluating text-to-visual generation with image-to-text generation[C]//European Conference on Computer Vision. Springer, Cham, 2025: 366-384.

---

> > > ### Author Response · Authors · 2024-11-21
> > > **(3) Official Comment to Reviewer dPkn**
> > >
> > > ## Q4. Implementation and results of comparative methods
> > >
> > > **We reproduced all baseline methods strictly using their publicly available codebases and parameter settings.** Since our focus is on **incomplete multi-view data**, we generated random missing masks, which may have introduced variations compared to the original papers. We followed the mask generation method from the **DSIMVC** repository (ICML '22 [9]) and applied the **same missing patterns and random seed** across all methods for consistency, as detailed in **Appendix C.1**.
> > >
> > > Regarding **Completer**, its original implementation is designed for two-view data. To adapt it to multi-view datasets, we followed the extension approach used in prior work (TPAMI '23 [3]). **Specifically, we evaluated all possible two-view combinations and reported the best clustering score**, ensuring a fair and meaningful comparison in the multi-view context. This process, along with consistent application to other two-view methods, is described in **Appendix D**.
> > >
> > > We hope this clarifies our implementation and ensures the transparency of our comparisons.
> > >
> > > [9] Tang H, Liu Y. Deep safe incomplete multi-view clustering: Theorem and algorithm[C]//International Conference on Machine Learning. PMLR, 2022: 21090-21110.
> > >
> > > ## Q5. Convergence Analysis of the loss function
> > >
> > > Your suggestion to include a convergence analysis greatly enriched our work. In response, we conducted both theoretical and experimental analyses:
> > > ### 1. **Theoretical Analysis**:
> > > - In **Appendix A.4**, we prove that the loss function serves as a valid evidence lower bound (ELBO) for the log marginal likelihood of incomplete multi-view data.
> > > - In **Theorem 1**, we demonstrate that the selected prior ensures the regularization term acts as a meaningful similarity measure across distributions, thereby supporting the optimization process.
> > > ### 2. **Experimental Analysis**:
> > > - We have included **Loss Evolution Curves** in **Figure 7**, which illustrate the progression of individual loss components during training on the Handwritten dataset with a missing rate of  $\eta=0.1$. These curves empirically confirm convergence:
> > >     - The **overall loss** decreases steadily and flattens over time, indicating stabilization. This decrease is primarily driven by the **Reconstruction Loss**, which initially focuses on self-view reconstruction by optimizing **Eq. (3)** during the first 100 epochs and then transitions to incorporate both self-view reconstruction and cross-view generation using **Eq. (4)** after 100 epochs.
> > >     - The **KLD_z term** initially rises and then declines, eventually stabilizing at a small value. This trend reflects the inter-view correspondences being progressively established in the latent space, as per the similarity measure defined in our method.
> > >     - The **KLD_$\omega$ term** also exhibits a similar trend of rising before falling, though its decline is less pronounced. This is due to the competing objectives of enforcing consistency in the fused latent representation ($\omega$) and retaining information for reconstruction.
> > > - To further validate convergence, we have included **latent space evolution visualizations** in **Appendix A.5**, showcasing how the latent variables $z$ and $\omega$ evolve during training. These visualizations illustrate **the successful alignment of multi-view latent representations and the establishment of inter-view correspondences.**
> > >
> > > This comprehensive analysis highlights the stability and effectiveness of our method, addressing your insightful suggestion.
> > >
> > > Thank you again for your detailed and constructive comments. We believe your feedback has significantly strengthened our manuscript, and we are confident that the revisions address the points you raised.

---

> > > ### Comment · Reviewer_dPkn · 2024-11-24
> > > **Thanks for your feedback**
> > >
> > > All my questions have been addressed, thanks

---

### Official Review · Reviewer_o4Jy · 2024-11-04

**Soundness:** 4
**Presentation:** 3
**Contribution:** 3
**Rating:** 8
**Confidence:** 3

**Summary:**

The paper introduces a novel model, Multi-View Permutation of Variational Autoencoders (MVP), designed to address incomplete multi-view data by establishing robust inter-view relationships through cyclic permutations within the VAE latent space. MVP uniquely integrates cyclic permutations and latent variable partitions to encode both shared and view-specific information, enabling inference for missing views without sacrificing inter-view coherence. By incorporating an informational prior through cyclic permutations, MVP transforms the regularization term into a similarity measure, enhancing the consistency and sufficiency of representations across views. Experimental results on seven benchmark datasets demonstrate that MVP outperforms existing IMVRL methods in clustering and generation tasks, showcasing its ability to generate coherent and robust representations even with high missing-view rates.

**Strengths:**

The application of cyclic permutations in the VAE latent space presents a unique and innovative approach to modeling inter-view relationships. This method appears more robust than prior approaches, effectively capturing and maintaining consistency across incomplete multi-view data.

MVP is rigorously tested across multiple datasets under varying missing rates, demonstrating strong adaptability and consistently superior performance over previous methods in both partially and fully observed data settings. The results confirm MVP’s effectiveness in handling diverse levels of data incompleteness.

The paper includes thorough experiments and analyses, such as ablation studies and additional tests on relatedness, offering comprehensive insights into MVP’s performance. The authors strengthen their findings by running models multiple times with different random seeds and presenting deviations on the plots, adding credibility to the robustness and reliability of their results.

**Weaknesses:**

Although the paper centers on incomplete multi-view learning, much of the processing related to incomplete data, such as section C1, is relegated to the appendix. This structure may hinder readability and comprehension. A more detailed description of the incomplete data processing steps in the main text would improve accessibility and clarity for readers.

While the proposed method’s effectiveness in handling random arrangements of latent variables is supported by experiments, it also introduces additional computational complexity, particularly in the cumulative computation of the multi-view variational lower bound function. MVP’s complexity seems closer to that of MMVAE and notably higher than MVAE and MVTCAE. A detailed analysis of this computational cost would be beneficial, as it has important implications for scalability in multi-view applications.

**Questions:**

Did the authors employ an analytical form or a sample-based estimation for the KL divergence terms in the ELBO computation?

How does the computational complexity of MVP scale with dataset size, and are there specific optimizations implemented to mitigate the computational overhead introduced by cyclic permutations?

---

> ### Author Response · Authors · 2024-11-21
> **Official Comment to Reviewer o4Jy**
>
> We sincerely appreciate your valuable feedback, which has greatly contributed to improving the clarity and robustness of our work. Below, we address each of your insightful comments in detail:
> ## Q1. Incomplete Data Processing Clarification
>
> Thank you for pointing out the importance of providing more clarity on the incomplete data processing steps. We have moved the description of the incomplete data processing to the beginning of the experimental section to improve readability and comprehension. Additionally, we have added a direct reference to Appendix C1 for further details. The revised content can be found in **lines 318-322, with changes highlighted in blue for your convenience**.
> ## Q2. Computational Complexity of MVP
>
> Your suggestion to provide a computational complexity comparison was incredibly helpful in clarifying the trade-offs of our approach. We have included a detailed comparison between our method and several baselines, as shown below:
>
> |                    | MVAE   | **MMVAE** | **mmJSD** | **MoPoE** | **MVTCAE** | **Ours**    |
> | ------------------ | ------ | --------- | --------- | --------- | ---------- | ----------- |
> | **Encoding**       | $O(V)$ | $O(V)$    | $O(V)$    | $O(2^V)$  | $O(V)$     | $O(V)$      |
> | **Transformation** | /      | /         | /         | /         | /          | $O(V(V-1))$ |
> | **Decoding**       | $O(V)$ | $O(KV^2)$ | $O(V)$    | $O(V)$    | $O(V)$     | $O(V)$      |
>
> Here, $V$ denotes the number of views, and $K$ is the number of samples of latent variables per view for optimizing the IWAE bound. It is worth noting that our method introduces additional computational cost during the transformation step because it explicitly constructs channels between views in the latent space. However, these transformations are performed in the low-dimensional latent space using a lightweight MLP, which keeps the added complexity manageable.
>
> We hope this comparison addresses your concern, and we thank you for encouraging us to include this analysis.
>
> ## Q3. Strategy of implementing Cyclic Permutations
>
> Your observation regarding the computational efficiency of cyclic permutations prompted us to elaborate further on the optimizations implemented in our method. As detailed in **Appendix C.2**, we adopted the following strategies to mitigate overhead:
>
> 1. **Pre-computation of Cyclic Permutation Indices**
>     - Using Sattolo's Algorithm, all cyclic permutations of $V$ indices can be efficiently generated with a time complexity of $O(V)$ for each permutation, resulting in $(V−1)!$ permutations in total. These permutations are precomputed and stored in a "fingerprint" file, along with pre-defined masks for incomplete multi-view datasets. By treating missing views as fixed points, we ensure that the permuted index sets retain consistent lengths, simplifying downstream operations.
> 2. **Batch Processing with Pre-stored Indices**
>     - During training, the precomputed indices are directly applied via indexing operations, enabling efficient cyclic permutations of latent variables (e.g., rearranging $Z_0$​ into $Z_1$​). This transformation is performed in batches using $L \times L$ arrays, ensuring seamless integration with the training pipeline.
>
> By trading a small amount of storage space for computational efficiency, our approach eliminates repetitive computations, ensuring scalability even with multiple views. We hope this explanation provides sufficient clarity on how we achieve computational efficiency.
> ## Q4. KL Divergence in ELBO Computation
>
> We appreciate your inquiry about the KL divergence terms in the ELBO computation. We used an **analytical form** for the KL divergence, as the terms involving $z$ and $\omega$ can be fully decoupled, as shown in Appendix A.4. Specifically, both the posterior and prior distributions in each KL term are multivariate Gaussians of the same dimensionality, which allows us to compute the KL divergence directly without any approximation.
>
> Thank you once again for your thoughtful and constructive comments, which have been instrumental in refining our manuscript. We are confident that the revisions based on your feedback have significantly improved the quality of our work.

---

> > ### Comment · Reviewer_o4Jy · 2024-11-26
> > **Reply to the authors**
> >
> > I appreciate the authors' response and will raise my score after reviewing the rebuttal.

---

> > > ### Author Response · Authors · 2024-11-27
> > > **Thank you for your feedback and score adjustment**
> > >
> > > We greatly appreciate your kind feedback and the score improvement. Your thoughtful comments have been invaluable, and we will continue to refine the paper during the final revision. Thank you again for your time and consideration!

---

### Author Response · Authors · 2024-11-21
**General Comment**

We would like to express our heartfelt gratitude to all reviewers for their thoughtful and detailed feedback. Your constructive comments and suggestions have been invaluable in refining our work, and we deeply appreciate the time and effort you have dedicated to reviewing our submission.

We are especially grateful for the recognition of the strengths of our work, as highlighted below:
1. **Importance of the Problem**:
	- “Missing is a common and significant problem in real-world multi-view learning.” by Reviewer dPkn
	- “The motivation is clearly described.” by Reviewer 7w7m
2. **Novelty of the Method**:
	- “A unique and innovative approach to modeling inter-view relationships with cyclic permutations.” by Reviewer o4Jy
	- “Applying permutations and segmentations in the latent space is very novel.” by Reviewer dPkn
	- “Successfully captures inter-view relationships and creates a robust latent space.” by Reviewer yAd5
 1. **Comprehensive Validation**:
	- “Rigorously tested across multiple datasets under varying missing rates.” by Reviewer o4Jy
	- “Comprehensive results on seven datasets, demonstrating robustness.” by Reviewer dPkn
	- “Compelling evidence of handling incomplete data effectively.” by Reviewer yAd5
	- “Extensive experiments demonstrate the methodology’s effectiveness.” by Reviewer 7w7m
	- “Thorough experiments with multiple seeds add credibility to the results.” by Reviewer o4Jy
4. **Clear Writing and Implementation Details**:
	- “Well-structured and easy to understand.” by Reviewer dPkn
	- “The paper is well-organized.” by Reviewer 7w7m
	- “Implementation details are presented in detail.” by Reviewer dPkn
	- “Provides rigorous theoretical analysis and proofs.” by Reviewer yAd5

We have carefully addressed all the constructive feedback provided by the reviewers. Specific revisions have been made to enhance the clarity, rigor, and reproducibility of our work. Updates have been incorporated into the revised manuscript, with all changes clearly highlighted in blue for your convenience.

We are confident that these improvements further strengthen our paper and look forward to hearing any additional thoughts you may have. Once again, we sincerely thank you for your detailed reviews and valuable insights.

---

### Meta-Review · Area_Chair_YPWX · 2024-12-20

**Metareview:**

This paper enhances the vanilla multi-modal VAE by enabling it to handle missing views through modeling inter-view correspondence. The idea of applying permutations and segmentations in the latent space to infer missing views is interesting, and the proposed approach is technically sound. The manuscript includes thorough theoretical analysis and empirical evaluations, both in the main text and the appendix, which strengthen the paper’s contribution. Three reviewers gave positive ratings (8, 6, 6), while one reviewer gave a borderline reject (5) but did not update the score after the rebuttal. Given that the authors addressed all the reviewer’s concerns in their response, I believe the reviewer would now be inclined to accept the paper. Overall, the paper is well-motivated, technically sound, and supported by solid empirical evaluations. I recommend acceptance.

**Additional Comments On Reviewer Discussion:**

Reviewers raised questions regarding the time complexity, approach details, and generalization. During the rebuttal stage, the authors provided detailed responses, including additional empirical results, that addressed most of these concerns.

---

### Decision · Program_Chairs · 2025-01-22

Accept (Poster)